# VisionLLM: Large Language Model is also an Open-Ended Decoder for Vision-Centric Tasks

**Wenhai Wang**[*2]  **Zhe Chen**[*1,3]  **Xiaokang Chen**[*1,4]  **Jiannan Wu**[*1,5]  **Xizhou Zhu**[1,6]
**Gang Zeng**[4]  **Ping Luo**[5]  **Tong Lu**[3]  **Jie Zhou**[6]  **Yu Qiao**[1]  **Jifeng Dai**[†1,6]
[1]OpenGVLab, Shanghai AI Laboratory  [2]The Chinese University of Hong Kong
[3]Nanjing University  [4]Peking University  [5]The University of HongKong  [6]Tsinghua University

Code: https://github.com/OpenGVLab/VisionLLM

## Abstract

Large language models (LLMs) have notably accelerated progress towards artificial general intelligence (AGI), with their impressive zero-shot capacity for user-tailored tasks, endowing them with immense potential across a range of applications. However, in the field of computer vision, despite the availability of numerous powerful vision foundation models (VFMs), they are still restricted to tasks in a pre-defined form, struggling to match the open-ended task capabilities of LLMs. In this work, we present an LLM-based framework for vision-centric tasks, termed VisionLLM. This framework provides a unified perspective for vision and language tasks by treating images as a foreign language and aligning vision-centric tasks with language tasks that can be flexibly defined and managed using language instructions. An LLM-based decoder can then make appropriate predictions based on these instructions for open-ended tasks. Extensive experiments show that the proposed VisionLLM can achieve different levels of task customization through language instructions, from fine-grained object-level to coarse-grained task-level customization, all with good results. It's noteworthy that, with a generalist LLM-based framework, our model can achieve over 60% mAP on COCO, on par with detection-specific models. We hope this model can set a new baseline for generalist vision and language models. The code shall be released.

## 1 Introduction

The emergence of large language models (LLMs) like ChatGPT [35] has revolutionized the landscape of artificial general intelligence (AGI), showcasing their impressive zero-shot capabilities in addressing various natural language processing (NLP) tasks through user-tailored prompts or language instructions. Despite these advancements, it's essential to note that the triumph of LLMs does not effortlessly extend to pure vision and vision-language tasks, due to the inherent disparities between modalities and task formats.

The field of computer vision presents a unique set of challenges and paradigms that differ from those of NLP. The traditional paradigm of vision foundation models is pre-training followed by fine-tuning [51, 11, 43, 53, 17, 44], which is effective but comes with significant marginal costs when adapting to diverse downstream scenarios. As shown in Figure 1a, while approaches such as multi-task unification [38, 50, 1, 49, 72] have been used to achieve generalist capability, they often struggle to overcome the limitations imposed by pre-defined tasks, resulting in a gap in open-ended task capabilities compared to LLMs. Recently, visual prompt tuning [24, 66, 70, 67, 54] has emerged as a way to flexibly outline some pure vision tasks (see Figure 1b), such as object detection, instance

---

* equal contribution, † corresponding author (daijifeng@tsinghua.edu.cn)

37th Conference on Neural Information Processing Systems (NeurIPS 2023).

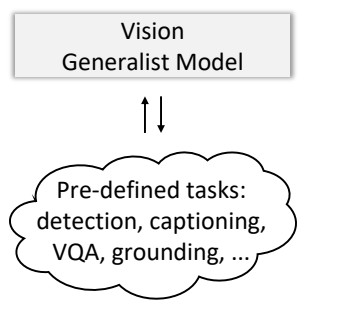
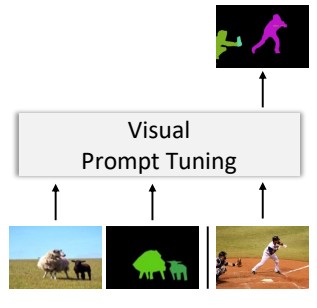
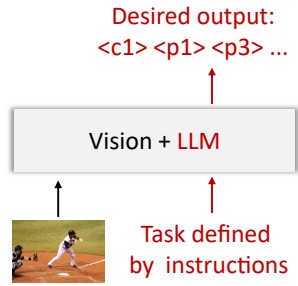

(a) Vision generalist models [51, 53, 74] are constrained by the format of pre-defined tasks.

(b) Visual prompt tuning [24, 56, 54] are inconsistent with the format of LLMs.

(c) VisionLLM (ours) can *flexibly manage vision-centric tasks using language instructions like LLMs*.

Figure 1: **Comparison of our VisionLLM with popular paradigms.** Unlike current vision generalist models that depend on pre-defined task formats and visual prompt tuning models that are inconsistent with large language models (LLMs), VisionLLM leverages the power of LLMs for open-ended vision tasks by using language instructions.

segmentation, and pose estimation, using visual masking. However, the format of visual prompts considerably deviates from that of language instructions, making it challenging to directly apply the reasoning abilities and world knowledge of LLMs to vision tasks. Therefore, *there is an urgent need for a unified generalist framework that can seamlessly integrate the strengths of LLMs with the specific requirements of vision-centric tasks.*

In this work, we present VisionLLM, a novel framework that aligns the definitions of vision-centric tasks with the methodologies of LLMs. Leveraging the reasoning and parsing capacities of LLMs, VisionLLM is designed to empower open-ended task capabilities for vision-centric tasks. Specifically, it comprises three core components: (1) a unified language instruction designed for vision and vision-language tasks, (2) a language-guided image tokenizer, and (3) an LLM-based open-ended task decoder that orchestrates various tasks using language instructions. With this framework, a wide range of vision-centric tasks can be seamlessly integrated, including object detection, instance segmentation, image captioning, and visual grounding. In addition, the framework also facilitates task customization at different levels of granularity, allowing for the customization of target objects, output formats, task descriptions, etc.

Compared to current popular API-based applications [60, 65, 42, 32, 28], our model takes a unified, end-to-end approach to integrate VFMs and LLMs, streamlining and enhancing the overall efficiency of the overall process, and leveraging the strengths and data of both VFMs and LLMs within a single, cohesive system. Furthermore, our model surpasses the limitations of generalist vision models pre-trained on pre-defined tasks. VisionLLM can effectively manage vision-centric tasks through language instructions, embodying a flexible and open-ended approach that is not constrained by pre-set tasks. This versatility makes VisionLLM a robust and powerful generalist model for vision and vision-language tasks, opening up new possibilities for the development of unified generalist models that bridge the domains of vision and language.

In summary, our main contributions are as follows:

(1) We propose VisionLLM, the first framework that leverages the power of LLMs to address vision-centric tasks in an open-ended and customizable manner. By aligning the definitions of vision-centric tasks with LLM methodologies, VisionLLM breaks new ground in enabling the unified modeling of vision and language, opening up possibilities for advancing the field.

(2) We overcome many difficulties when porting LLMs to vision-centric tasks, by designing unified language instruction that matches the format of language models and covers various vision-centric tasks including visual perception. Correspondingly, we develop a language-guided image tokenizer and an LLM-based task decoder that can handle open-ended tasks according to the given language instructions based on the LLMs' reasoning and parsing capabilities.

(3) We construct a series of tasks with different granularities to verify the effectiveness of our models, ranging from easy to hard, and from pre-defined to flexible. Through these validations, we demonstrate the remarkable generality of our models, showcasing their ability to handle diverse

**Human:** "*Identify the objects in* <image> *that belong to {'What is the child eating?':* <c0>, *'red gamepad':* <c1>*} and draw a bounding box around each one. The output should be a list of tuples in*

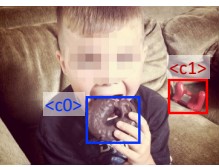

*the format (c, x1, y1, x2, y2), where c is the class label and x1, y1, x2, y2 are the coordinates of the top-left and bottom-right corners of the bounding box. The coordinates should be within the range 1024*."

**VisionLLM:** "*The bounding boxes are [(<c0>, 226.4, 229.8, 363.1, 347.4), (<c1>, 441.1, 183.5, 538.6, 269.9)]*."

(a) Object Detection with *Customized Class Set*

**Human:** "*For each object in* <image> *that is a member of {'motorbike':* <c0>*}, output a tuple with the class label and the coordinates of a polygon with 16 points that*

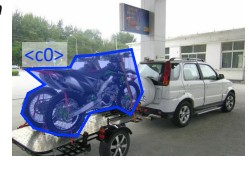

*encloses the object. The coordinates should be within the range 1024. The output format should be (c, x1, y1, x2, y2, ..., x16, y16)*."

**VisionLLM:** "*The detected polygons are [(<c0>, 135.3, 95.7, 123.4, 53.4, 84.9, 57.6, 66.8, 60.5, 60.1, 72.3, 34.2, 71.4, ..., 124.9, 119.3)]*."

(b) Object Location with *Editable Output Format*

**Human:** "*Provide a long description of the* <image>."

**VisionLLM:** "*The image features a white dog laying down on a foldable outdoor cooling pad ,*

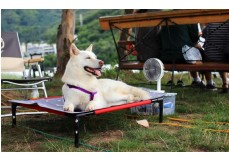

*which is placed on top of a wooden table. The dog has its head in the middle of the cooling pad, and a fan is positioned behind it, blowing air onto the pad. The table is outdoors, in a grassy area, and there are two chairs placed around it, one on each side. The scene appears to be set in a sunny weather, with a bright and sunny sky, which contributes to the overall outdoor atmosphere*."

(c) Image Description with *Controllable Text Length*

**Human:** "*With the objects in the* <image>, *if you feel hot, what will you do?*"

**VisionLLM:** "*First, You can also try turning off the computer and other electrical appliances in the*

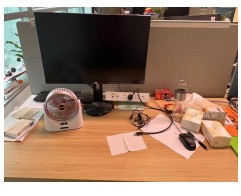

*room to reduce the heat generated by their operation. As an alternative, you can use the desk fan provided in the image to create a cooling effect in the room. You can plug in the fan and adjust its settings to ensure it blows air in a direction that reaches the room's occupants effectively.*"

(d) Visual Question Answer with *Complex Reasoning*

Figure 2: **Results and visualizations of our VisionLLM**. Guided by language instructions, our unified generalist framework showcases its effectiveness on diverse open-ended vision-centric tasks. The text marked with a gray background indicates the customized instructions and the desired outputs.

scenarios, including random object categories, random output formats, and random task descriptions, as shown in Figure 2. The successful outcomes of these validations underscore the tremendous potential of our model in harnessing the capabilities of LLMs to control and guide vision-centric tasks. In addition, with a generalist LLM-based framework, our model also yields promising results on various vision-centric tasks. Notably, our generalist model achieves an impressive mAP score of 60+% on the COCO dataset, surpassing many detection-specific models [73, 6, 20] and approaching the state-of-the-art record.

## 2 Related Work

### 2.1 Large Language Model

Large language models (LLMs) have gained significant attention in the field of natural language processing (NLP) and artificial general intelligence (AGI), due to their impressive capabilities in language generation, in-context learning, world knowledge, and reasoning. The GPT family, including GPT-3 [5], ChatGPT [35], GPT-4 [34], and InstructGPT [36] are most representative works of LLMs. Other LLMs like OPT [69], LLaMA [46], MOSS [14], and GLM [68] have also made substantial contributions to the field. These models achieve high performance and are open-sourced, serving as valuable resources for training large models and as foundations for further fine-tuning for specific purposes. For instance, Alpaca [45] introduces a self-instruct framework that facilitates instruction tuning of the LLaMA model, reducing the reliance on human-written instruction data. Recently, the emergence of these LLMs has also opened up API-based applications for solving vision-centric tasks. These applications have integrated visual APIs with language models to enable decision-making or planning based on visual information, such as Visual ChatGPT [60], MM-REACT [65], HuggingGPT [42], InternGPT [32], and VideoChat [28]. However, despite the convenience of using language-based instructions to define tasks and describe visual elements, these interactive systems [60, 65, 42, 32, 28] still face limitations in capturing fine-grained visual details and understanding complex visual contexts, which hinder their ability to effectively connecting vision

and language models. In summary, while LLMs have shown tremendous potential in various NLP applications, their applicability to vision-centric tasks has been limited by the challenges posed by modalities and task formats.

## 2.2 Vision Generalist Model

The pursuit of generalist models [74, 33, 62], which aim to handle a wide range of tasks using a shared architecture and parameters, has been a long-standing goal in the machine learning community. Inspired by the success of sequence-to-sequence (seq2seq) models in the field of NLP [38], recent advancements such as OFA [50], Flamingo [1], and GIT [49] propose modeling diverse tasks as sequence generation tasks. Unified-IO [33], Pix2Seq v2 [8], and UniTab [63] extend this idea by using discrete coordinate tokens to encode and decode spatial information for more tasks. Gato [39] also incorporates reinforcement learning tasks into the seq2seq framework, while GPV [19] develops a general-purpose vision system by combining a seq2seq module with a DETR-based visual encoder [6]. However, these methods suffer from some limitations, such as slow inference speed and performance degradation due to the non-parallel auto-regressive decoding process. Uni-Perceivers [74, 72, 26] solve these issues by unifying different tasks using the maximum likelihood target for each input based on representation similarity, regardless of their modality, making it possible to support both generation and non-generation tasks in a unified framework. Nevertheless, these generalist models are still restricted by pre-defined tasks and cannot support flexible open-ended task customization based on language instructions like LLMs.

## 2.3 Instruction Tuning

Language instructions are a powerful way to express various NLP tasks and examples for LLMs, as introduced by GPT-3 [5]. Following this idea, subsequent works, such as InstructGPT [36], FLAN [13, 59], and OPT-IML [23], explore the instruction-tuning method [58, 57] and demonstrate that this simple approach effectively enhances the zero-shot and few-shot capabilities of LLMs. The language instruction paradigm has also been adopted by the computer vision community to define image-to-text tasks. Flamingo [1] is a milestone work that uses vision and language inputs as prompts and achieves remarkable few-shot results in various vision-language tasks, such as image captioning [9] and VQA [2]. BLIP-2 [27] further connects the visual encoder with LLMs through a querying transformer and a linear projection layer to build strong multimodal models. MiniGPT-4 [71] and LLaVA [30] finetune the BLIP-2-style models on synthetic multimodal instruction-following data to unleash the potential of LLMs. However, these models mainly focus on image-to-text tasks and fail to address visual perception, such as object detection, instance segmentation, pose estimation, etc. To tackle image inpainting tasks, Bar *et al.* [3] introduces the first visual prompting framework that utilizes inpainting with discrete tokens on images. Painter [55] and SegGPT [56] employ masked image modeling on raw pixels for in-context learning with paired images. While these visual prompt models demonstrate good results in segmentation tasks, their applicability to numerous real-world vision tasks is challenging. Moreover, defining the visual prompts as image inpainting is inconsistent with the language instructions in LLMs, hard to leverage the reasoning, parsing ability, and world knowledge of LLMs. In this work, we aim to align vision-centric tasks with language tasks, use language instructions to unifiedly and flexibly define all tasks, and solve them with a shared LLM-based task decoder.

# 3 VisionLLM

## 3.1 Overall Architecture

This work targets to provide a unified generalist framework that can seamlessly integrate the strengths of large language models (LLMs) with the specific requirements of vision-centric tasks. As shown in Figure 3, the overall architecture of VisionLLM consists of three key designs: (1) a unified language instruction that provides a consistent interface for vision-centric task definition and customization; (2) a language-guided image tokenizer, which encodes visual information in alignment with the given language prompt, enabling the model to comprehend and parse the visual content effectively; and (3) an LLM-based open-task decoder, which utilizes the encoded visual information and language instructions to generate satisfactory predictions or outputs. The three designs work together to achieve a flexible and open-ended framework that can handle various vision-centric tasks at different levels of task customization through language instructions.

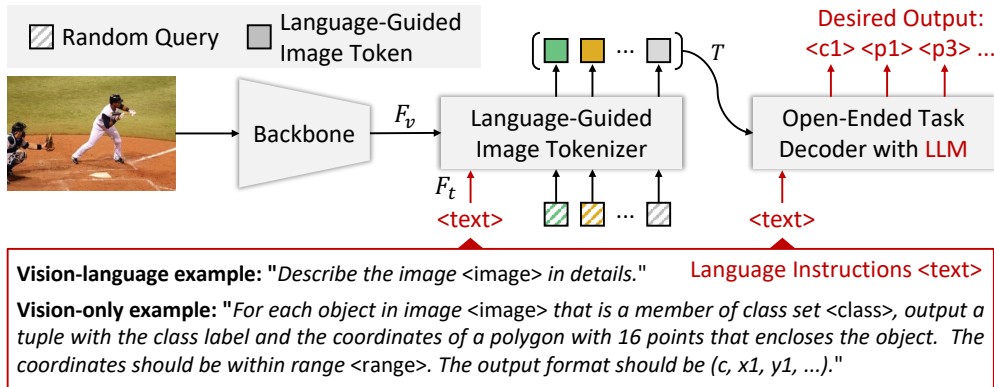

Figure 3: **Overall architecture of the proposed VisionLLM.** It consists of three parts: a unified language instruction designed to accommodate both vision and vision-language tasks, an image tokenizer that encodes visual information guided by language instructions, and an LLM-based open-ended task decoder that executes diverse tasks defined by language instructions.

Different from previous interactive systems [60, 65, 42, 32, 28] that rely on APIs, our VisionLLM presents a more flexible and end-to-end pipeline. Given language instructions that describe the current tasks and an input image, the model first uses a language-guided image tokenizer to encode the image tokens based on the given prompt. Then, the image tokens and language instructions are fed to an LLM-based open-ended task decoder. Finally, it evaluates the generated outputs against the task definition given by the unified language instructions, enabling the model to produce task-specific results. This seamless, end-to-end pipeline enables VisionLLM to effectively combine vision and language, achieving remarkable performance in open-ended and customizable vision-centric tasks.

## 3.2 Unified Language Instruction

We first introduce unified language instructions to describe vision-centric tasks. This design enables the unification of various vision-only and vision-language task descriptions and allows for flexible task customization.

**Vision-Language Tasks.** The instructions for vision-language tasks such as image captioning and visual question answering (VQA) are straightforward and similar to NLP tasks. Following previous methods [27, 74, 30], we describe the image captioning task like "*The image is* `<image>`. *Please generate a caption for the image:* ", and the VQA task like "*The image is* `<image>`. *Please generate an answer for the image according to the question:* `<question>`". Here, `<image>` and `<question>` are the placeholders of the image tokens and the question, respectively. The image tokens are directly placed at the placeholder `<image>`.

**Vision-Only Tasks.** Designing effective language instructions for vision tasks is a challenging endeavor due to the differences in modality and task format between vision and language. Here, we describe vision tasks by providing a task description and specifying the desired output format via language instructions.

(1) The task description conveys the intended task to the language model. Following self-instruct [57], we design a set of seed instructions with placeholders and employ LLMs to generate a large number of related task descriptions and randomly select one of them during training.

(2) For conventional visual perception tasks like object detection and instance segmentation, we propose a unified output format represented as a tuple $(C, P)$, where $C$ denotes the class index in the category set `<class>`, and $P = \{x_i, y_i\}_{i=1}^{N}$ represents $N$ points that locate the object. To align with the format of word tokens, both the class index $C$ and the coordinates of points $x_i, y_i$ are transformed into discretized tokens. Specifically, the class index is an integer starting from 0, and the continuous coordinates of the points are uniformly discretized into an integer within the range [-`<range>`, `<range>`]. For object detection and visual grounding tasks, the point number $N$ is equal to 2, representing the the top-left and bottom-right points of object's bounding box. In the case of instance segmentation, we employ multiple ($N > 8$) points along the object boundary to represent an instance mask [61]. Other perception tasks such as pose estimation (keypoint detection) can also be formulated as language instructions in this way.

An example of language instruction for the instance segmentation task is as follows: "*Segment all the objects of category set `<class>` within the `<range>` of the image and generate a list of the format (c, x1, y1, x2, y2, ..., x8, y8). Here, c represents the index of the class label starting from 0, and (x1, y1, x2, y2, ..., x8, y8) correspond to the offsets of boundary points of the object relative to the center point. The image is:* `<image>`".

### 3.3 Language-Guided Image Tokenizer

VisionLLM considers images as a kind of foreign language and converts them into token representations. Unlike previous works [16, 52, 31] that utilize fixed-size patch embeddings to represent images, we introduce the language-guided image tokenizer to flexibly encode visual information that aligns with task-specific language prompts or instructions.

Specifically, give an image $\mathbf{X} \in \mathbb{R}^{H \times W \times 3}$ with height $H$ and width $W$, we first feed it to the image backbones (*e.g.*, ResNet [21]) and extract visual features $F_v$ of four different scales. Additionally, we leverage a text encoder (*e.g.*, BERT [15]) to extract the language features $F_l$ from given prompts. The language features are then injected into each scale of visual features through cross-attention [47], yielding multi-scale language-aware visual features, enabling the alignment of features across modalities.

Afterward, we propose to adopt a transformer-based network (*e.g.*, Deformable DETR [73]) with $M$ random-initialized queries $Q = \{q_i\}_{i=1}^{M}$ to capture the high-level information of images. We build the transformer-based network on top of the multi-scale language-aware visual features to extract $M$ image tokens $T = \{(e_i, l_i)\}_{i=1}^{M}$, each of which is represented by an embedding $e_i$ and a location $l_i$, denoting the semantic and positional information of the token. This design not only represents the images independent of input resolution but also extracts the visual representation that is informative with respect to the language prompts.

### 3.4 LLM-based Open-Ended Task Decoder

We build our decoder on Alpaca [45], an LLM that is adapted from LLaMA [46], to handle various vision-related tasks with language guidance. However, Alpaca has some inherent drawbacks for vision-centric tasks, such as (1) It only has a few digit tokens (*e.g.*, 0∼9) in its vocabulary, which restricts its ability to locate objects by numbers; (2) It uses multiple tokens to represent the category name, resulting in an inefficient scheme in object classification; and (3) It is a causal model that is inefficient for visual perception tasks.

To tackle these issues, we expand the vocabulary of LLM with additional tokens specially designed for vision-centric tasks. First, we add a set of location tokens, denoted as {`<p-512>`, ..., `<p0>`, ..., `<p512>`}, where `<p i>` represents the discretized offset of $i \in [-512, 512]$ to the location $l_i$ of the image token, and the relative value to image height or width is equal to $i/512$. These tokens successfully transform the object localization task from continuous variable prediction to more unified discrete bin classification. Second, we introduce semantics-agnostic classification tokens {`<c0>`, `<c1>`, ..., `<c511>`} to replace category name tokens, which overcomes the inefficiency of using multiple tokens to represent categories. The mapping between category names and the classification tokens is flexibly provided in the category set `<class>` of language instructions, such as {`"person":<c0>`, `"car":<c1>`, `"black cat":<c2>`,...}. This design allows our model to select the appropriate category name from the provided category set, facilitating efficient and accurate object classification.

Moreover, to address the inefficiency caused by the causal framework, we introduce output-format-as-query decoding. We first use LLMs to parse the structural output format from the task instructions (*e.g.*, "`<cls> <x1> <y1> <x2> <y2>`" for object detection, "`<bos>`" for image captioning), and then feed the tokens of structural output format as queries to the decoder to generate the desired output according to the queries. This simple method enables our model to not only avoid inefficient token-by-token decoding in visual perception tasks, but also keep a unified framework for vision-language tasks.

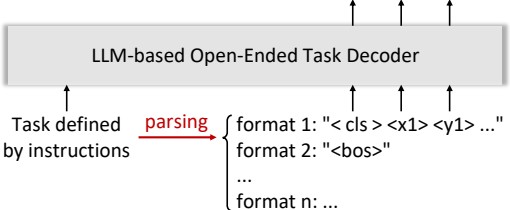

Figure 4: Illustration of the "output-format-as-query" decoding process. "`<cls> <x1> <y1> ...`" denote the queries of the object's class index and boundary points, and "`<bos>`" denotes the beginning of string.

Note that, during both the training and inference phases in the object detection task, we input 100 sets of "<cls> <x1> <y1> <x2> <y2>" to the decoder, generating 100 object predictions. Those predictions with higher confidence scores will be retained, adhering to a common practice of the object detection task.

In this way, the output of object location and classification is formulated as a foreign language, thus unifying these vision-centric tasks into the format of token classification. Therefore, both vision-language and vision-only tasks can be supervised with the cross-entropy loss like language tasks. In addition, for efficient training, we adopt the Low-Rank Adaptation (LoRA) approach [22], which allows us to train and fine-tune the models without excessive computational costs. We set the LoRA rank to 64 and use LoRA on the QKVO (Query, Key, Value, and Output) in the attention layers. It also acts as a bridge between the language and visual tokens, facilitating effective alignment between the two modalities, ensuring better task customization, and improving the convergence of the overall system.

# 4 Experiment

## 4.1 Implementation Details.

We implement two variants of VisionLLM with two image backbones, *i.e.*, ResNet [21] and InternImage-H [51]. For the language-guided image tokenizer, we adopt BERT-Large [4] as the text encoder and Deformable DETR (D-DETR) [73] to capture high-level information. For the LLM, we employ Alpaca-7B [45], a LLaMA [46] model fine-tuned with instructions, and equip it with LoRA [22] for parameter-efficient fine-tuning.

The model is trained in two stages. In the first stage, we initialize the model with the pre-trained weights of D-DETR and BERT, and train the visual backbone and language-guided image tokenizer to produce language-aware visual features. In the second stage, we connect the image tokenizer with Alpaca-7B and introduce the unified supervision of multiple tasks. We freeze the visual backbone while freezing most parameters of the LLM except a few LoRA parameters. More details on the experimental setup can be found in Sec. B of the supplementary material.

## 4.2 Task-Level Customization

We first evaluate the task-level customization capability of VisionLLM. VisionLLM supports coarse-grained task customization, including visual perception tasks and visual-language tasks. Table 1 presents the evaluation results on four standard vision-centric tasks, including object detection, instance segmentation, visual grounding, and image captioning. We compare our model with task-specific methods as well as recently-proposed vision generalist models. Note that, unless specifically mentioned, *the results of our model come from a shared-parameter generalist model and switch different tasks by changing the language instructions only. Detailed instructions could be found in the supplementary material.*

**Object Detection.** Object detection is a fundamental computer vision task that involves identifying and localizing objects of interest within an image. Our method achieves comparable or higher results to others, $44.6$ mAP, with a ResNet-50 [21] backbone. With the same backbone *i.e.* ResNet-50, our method outperforms Pix2Seq [7] by $1.4$ mAP, which also discretizes the output coordinates to integers. Furthermore, benefiting from the output-format-as-query framework (see Sec. 3.4), we can decode multiple predictions in parallel during inference, making our approach more efficient. Using InternImage-H [51] as the visual backbone, we obtained 60.2% mAP, which is close to the current state-of-the-art detection-specific model [51], demonstrating the scalability of our generalist model.

**Visual Grounding.** Visual grounding associates textual descriptions with corresponding regions or objects within an image. Training visual grounding and object detection can potentially conflict with each other, as object detection aims to detect all the objects, while visual grounding should only localize the referred object and suppress other objects. Benefiting from our unified task instructions and the strong instruction comprehension capabilities of LLMs, our model performs both tasks effectively and achieves a result of $80.6$ P@0.5 for visual grounding. With InternImage-H as the backbone, we achieve $86.7$ P@0.5 on the validation set of RefCOCO.

**Instance Segmentation.** Instance segmentation involves identifying and segmenting individual objects within an image. We employ a flexible number of points (*i.e.*, 8~24) along the object boundary to represent an instance mask. Compared to mainstream models specific to instance

Table 1: **Results on standard vision-centric tasks.** "sep" indicates that the model is separately trained on each task.

| Method | Backbone | Open-Ended | Detection | | | Instance Seg. | | | Grounding | Captioning | |
|---|---|---|---|---|---|---|---|---|---|---|---|
| | | | AP | AP$_{50}$ | AP$_{75}$ | AP | AP$_{50}$ | AP$_{75}$ | P@0.5 | BLEU-4 | CIDEr |
| *Specialist Models* | | | | | | | | | | | |
| Faster R-CNN-FPN [40] | ResNet-50 | ✗ | 40.3 | 61.0 | 44.0 | - | - | - | - | - | - |
| DETR-DC5 [6] | ResNet-50 | ✗ | 43.3 | 63.1 | 45.9 | - | - | - | - | - | - |
| Deformable-DETR [73] | ResNet-50 | ✗ | 45.7 | 65.0 | 49.1 | - | - | - | - | - | - |
| Mask R-CNN [20] | ResNet-50 | ✗ | 41.0 | 61.7 | 44.9 | 37.1 | 58.4 | 40.1 | - | - | - |
| Polar Mask [61] | ResNet-50 | ✗ | - | - | - | 30.5 | 52.0 | 31.1 | - | - | - |
| Pix2Seq [7] | ResNet-50 | ✗ | 43.2 | 61.0 | 46.1 | - | - | - | - | - | - |
| UNITER [10] | ResNet-101 | ✗ | - | - | - | - | - | - | 81.4 | - | - |
| VILLA [18] | ResNet-101 | ✗ | - | - | - | - | - | - | 82.4 | - | - |
| MDETR [25] | ResNet-101 | ✗ | - | - | - | - | - | - | 86.8 | - | - |
| BEiT-3 [53] | ViT-g | ✗ | - | - | - | - | - | - | - | - | 147.6 |
| VL-T5 [12] | T5-B | ✗ | - | - | - | - | - | - | - | - | 116.5 |
| *Generalist Models* | | | | | | | | | | | |
| UniTab [64] | ResNet-101 | ✗ | - | - | - | - | - | - | 88.6 | - | 115.8 |
| Uni-Perceiver [74] | ViT-B | ✗ | - | - | - | - | - | - | - | 32.0 | - |
| Uni-Perceiver-MoE [72] | ViT-B | ✗ | - | - | - | - | - | - | - | 33.2 | - |
| Uni-Perceiver-V2 [26] | Swin-B | ✗ | 58.6 | - | - | 50.6 | - | - | - | 35.4 | 116.9 |
| Pix2Seq v2 [8] | ViT-B | ✗ | 46.5 | - | - | 38.2 | - | - | - | 34.9 | - |
| VisionLLM-R50$_{sep}$ | ResNet-50 | ✗ | 44.8 | 64.1 | 48.5 | 25.2 | 50.6 | 22.4 | 84.4 | 30.8 | 112.4 |
| VisionLLM-R50 | ResNet-50 | ✓ | 44.6 | 64.0 | 48.1 | 25.1 | 50.0 | 22.4 | 80.6 | 31.0 | 112.5 |
| VisionLLM-H | InternImage-H | ✓ | 60.2 | 79.3 | 65.8 | 30.6 | 61.2 | 27.6 | 86.7 | 32.1 | 114.2 |

Table 2: **Experiments of object-level and output format customization.** We conduct these experiments based on VisionLLM-R50, and report the performance of box AP and mask AP on COCO minival for (a) and (b), respectively. "#Classes" and "#Points" indicate the number of classes and boundary points, respectively. "*" indicates that we report the mean AP of the given classes, *e.g.*, 10 classes.

(a) Object-level customization.

| #Classes | AP | AP$_{50}$ | AP$_{75}$ | AP$_S$ | AP$_M$ | AP$_L$ |
|---|---|---|---|---|---|---|
| 10* | 48.9 | 72.6 | 51.2 | 31.7 | 47.5 | 67.3 |
| 20* | 52.7 | 73.6 | 56.8 | 31.8 | 53.2 | 70.5 |
| 40* | 49.3 | 70.7 | 53.2 | 33.1 | 53.6 | 63.8 |
| 80* | 44.6 | 64.0 | 48.1 | 26.7 | 47.9 | 60.5 |

(b) Output format customization.

| #Points | AP | AP$_{50}$ | AP$_{75}$ | AP$_S$ | AP$_M$ | AP$_L$ |
|---|---|---|---|---|---|---|
| 8 | 18.5 | 45.7 | 11.6 | 9.9 | 19.7 | 28.7 |
| 14 | 22.9 | 48.3 | 19.4 | 11.0 | 25.1 | 36.0 |
| 16 | 24.2 | 49.9 | 20.9 | 11.5 | 26.3 | 36.8 |
| 24 | 25.1 | 50.0 | 22.4 | 12.5 | 27.4 | 38.2 |

segmentation, our model has a comparable mask AP$_{50}$ (61.2% with InternImage-H [51]) but relatively low mask AP$_{75}$. This gap could potentially arise from factors as follows: (1) We discretize the output coordinates to integers for unifying tasks, which introduces information loss; (2) Due to the memory and computational constraint, the number of points in our model is limited, which also results in a performance drop; and (3) Point-based methods typically yield lower results compared to direct mask prediction methods, such as Mask R-CNN [20].

**Image Captioning.** We also evaluate our model in a representative vision-language task, *i.e.* image captioning task, and report the BLEU-4 [37] and CIDEr [48] metrics. Note that we do not adopt the CIDEr optimization [41]. We can observe that VisionLLM achieves competitive performance to previous methods. With ResNet-50, we obtain a BLEU-4 score of 31.0 and a CIDEr score of 112.5. When using InternImage-H as the backbone, our model achieves a comparable BLEU-4 score of 32.1 and a CIDEr score of 114.2. These results demonstrate the effectiveness of VisionLLM in generating descriptive and contextually relevant captions for images.

### 4.3 Object-Level & Output Format Customization

Our VisionLLM not only allows for customizing the task description, but also for adjusting the target object and the output format using language instructions. Here, we evaluate our model's fine-grained customization ability on COCO. In particular, to customize the target object, we modify the <class> in language instructions to change the model's recognition target from 10 classes to 80 classes. Likewise, to customize the output format, we modify the number of points in language instructions to change the task output format. Table 2 shows that our method can perform well for both object-level and output format changes.

Table 3: **Ablation studies on language-guided image tokenizer and hyper-parameters.**

(a) Effect of text encoder in the language-guided image tokenizer.

| w/ BERT | Freeze | COCO | RefCOCO |
|---------|--------|------|---------|
| - | - | 44.7 | 48.1 |
| ✓ | - | 44.8 | 84.1 |
| ✓ | ✓ | 1.3 | 34.3 |

(b) Effect of image tokenization method.

| Tokenization | AP |
|--------------|------|
| Average Pooling | 23.1 |
| Ours | 44.8 |

(c) Effect of the number of bins (#Bins).

| #Bins | AP |
|-------|------|
| 257 | 34.9 |
| 513 | 40.8 |
| 1025 | 44.8 |
| 2049 | 44.8 |

## 4.4 Ablation Study

In this section, we analyze the effect of key components and hyper-parameters on VisionLLM. Unless otherwise specified, we use ResNet-50 [21] backbone and perform the ablation experiments for object detection tasks with random classes and task descriptions on COCO2017 [29].

**Single Task *vs*. Multiple Tasks.** We perform an ablation study to assess the impact of multi-task learning with language instructions on VisionLLM. As shown in Table 1, the single-task trained model VisionLLM-R50$_{sep}$ is slightly better than the jointly trained model VisionLLM-R50 except image captioning. This is due to the multitasking conflicts that also affect previous generalist models [74, 72], and it reflects a trade-off between accuracy and generalization.

**Text Encoder in Language-Guided Image Tokenizer.** We examine the role of text encoder (*i.e.*, BERT) in our language-guided image tokenizer in Table 3a, where we report the results for object detection and visual grounding. The first two rows show that BERT is not essential for object detection but it is crucial for visual grounding. We also investigate the effect of freezing the text encoder during training. The last row indicates that freezing BERT hinders the alignment of vision and language modalities and thus degrades the performance for both tasks.

**Image Tokenization Method.** As a comparison to our query-based tokenization, we employ average pooling on the feature maps from the D-DETR encoder to obtain $M$ patch embeddings, which serve as token representations for the image. Results in Table 3b indicate a clear advantage of our method. This is due to its ability to capture information from objects of various sizes in a more flexible way.

**Number of Localization Tokens.** We vary the number of localization tokens from 257 (*i.e.*, -128∼128) to 2049 (*i.e.*, -1024∼1024), to investigate its impact on visual perception performance. As presented in Table 3c, the model consistently exhibits improvement as the number of localization tokens increases until it reaches a saturation point. Remarkably, a substantial performance boost is observed when the number is raised from 257 to 1025 (+9.9 AP). These results indicate that a higher number of localization tokens enables the models to achieve finer localization abilities, thereby improving localization accuracy.

## 5 Conclusion

In this paper, we have presented VisionLLM, a novel framework that leverages the power of large language models (LLMs) to address vision-centric tasks in an open-ended and customizable manner. We have designed unified language instruction that matches the format of language models and covers various vision-centric tasks including visual perception. We have also developed a language-guided image tokenizer and an LLM-based task decoder that can handle open-ended tasks according to the given language instructions. We have verified the effectiveness of our models on a series of tasks with different granularities, demonstrating their remarkable generality and flexibility.

**Broader Impact.** We envision that this work will promote the fusion of visual and language tasks. In addition, since our work is built on open-source pre-trained vision foundation models and large language models, requiring low training resources, thus reducing the carbon footprint. We do not foresee obvious undesirable ethical/social impacts at this moment.

## Acknowledgement

The work is supported by the National Key R&D Program of China (NO. 2022ZD0161300), the National Natural Science Foundation of China (Grant No. 62376134, 61672273, 61832008, 62372223), the Shanghai Committee of Science and Technology (Grant No. 21DZ1100100), and the Fundamental Research Funds for the Central Universities (No. XJ2023000701).

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
