# Supplementary Materials for "VisionLLM: Large Language Model is also an Open-Ended Decoder for Vision-Centric Tasks"

**Wenhai Wang**[*2]    **Zhe Chen**[*1,3]    **Xiaokang Chen**[*1,4]    **Jiannan Wu**[*1,5]    **Xizhou Zhu**[1,6]
**Gang Zeng**[4]    **Ping Luo**[5]    **Tong Lu**[3]    **Jie Zhou**[6]    **Yu Qiao**[1]    **Jifeng Dai**[†1,6]
[1]OpenGVLab, Shanghai AI Laboratory    [2]The Chinese University of Hong Kong
[3]Nanjing University    [4]Peking University    [5]The University of HongKong    [6]Tsinghua University

Code: https://github.com/OpenGVLab/VisionLLM

## A    Example Instructions

As described in Sec. 3.2 of the main paper, we follow self-instruct [25] to design a set of seed instructions with placeholders and employ LLMs to create diverse related task descriptions for coarse-grained task-level customization. Here, we show some examples of instructions for task-level customization, including object detection, instance segmentation, visual grounding, image captioning, and visual question answering (VQA). *Following various instructions, our model can elegantly switch among different vision-centric tasks and accomplish them in a unified manner like LLMs.*

### A.1    Object Detection

**Example 1.** "*Please examine the image and identify all objects in the category set* `<class>`. *For each object, specify its location within the range* `<range>` *by determining the top-left and bottom-right corners of its bounding box. To indicate the object's class and location, provide the output in the format (c, x1, y1, x2, y2), where 'c' represents the class index starting from 0, and (x1, y1, x2, y2) correspond to the offsets of the bounding box corners relative to the center point. The image is:* `<image>`"

**Example 2.** "*Identify all the objects in the image that belong to the category set* `<class>` *and predict a bounding box around each one. The output should be a list in the format (c, x1, y1, x2, y2), where c represents the index of the class label starting from 0, and x1, y1, x2, y2 are the offsets of the top-left and bottom-right corners of the box relative to the center point. The coordinates should be within* `<range>`. *The image is:* `<image>`"

**Example 3.** "*For each object in the image that is a member of the category set* `<class>`, *output a tuple with the index of class label starting from 0 and the offsets of corners relative to the center point that encloses the object. The offsets should be in the order of top-left and bottom-right corners of the rectangle and should be within* `<range>`. *The output format should be (c, x1, y1, x2, y2). The image is:* `<image>`"

### A.2    Instance Segmentation

**Example 1.** "*Segment the objects from the image with class labels from* `<class>` *and output their coordinates within range* `<range>`. *The coordinates should be given as the boundary points relative to the center point, and the output format should be (c, x1, y1, x2, y2, ..., x20, y20), where c is the index of the class label that starts from 0. The image is:* `<image>`"

37th Conference on Neural Information Processing Systems (NeurIPS 2023).

**Example 2.** "*Segment all the objects from the category set* `<class>` *in the provided image and output a tuple (c, x1, y1, x2, y2, ..., x14, y14) for each, where c is the index of the class label in the category set that starts from 0, and (x1, y1, x2, y2, ..., x14, y14) correspond to the offsets of boundary points on the instance mask relative to the center point which should be within* `<range>`. *The image is:* `<image>`"

**Example 3.** "*In the provided image, please segment all the objects in category set* `<class>` *within the range* `<range>` *by providing their coordinates in the (c, x1, y1, x2, y2, ..., x24, y24) format, where 'c' denotes the index of the class label starting from 0, and (x1, y1, x2, y2, ..., x24, y24) stand for the offsets of boundary points relative to the center point. The image is:* `<image>`"

### A.3   Visual Grounding

**Example 1.** "*Please find the object in the category set* {`<expression>`:`<cls0>`} *within the range* `<range>`. *Please provide the output in the format (c, x1, y1, x2, y2), where c is the class index starting from 0, and (x1, y1, x2, y2) are the offsets of the top-left and bottom-right corners of the bounding box relative to the center point. The image is:* `<image>`"

**Example 2.** "*Given the input image, category set* {`<expression>`:`<cls0>`}, *and the range* `<range>`, *please locate the object in the image and output the corresponding coordinates in the tuple (c, x1, y1, x2, y2), where c is the index of the class label starting from 0, and (x1, y1, x2, y2) are the offsets of the top-left and bottom-right corners of the rectangle relative to the center point. The image is:* `<image>`"

**Example 3.** "*For each object in the image that belongs to the* {`<expression>`:`<cls0>`} *category set, please provide the class label (starting from 0) and the offsets from the center of a bounding box that encloses the object. The corner offsets should be in the order of top-left and bottom-right, and within the range* `<range>`. *The output should be in the format (c, x1, y1, x2, y2). The image is:* `<image>`"

### A.4   Image Captioning

**Example 1.** "*The image is* `<image>`. *Write a caption:* "

**Example 2.** "*The image is* `<image>`. *Please describe this image:* "

**Example 3.** "*With the objects in the* `<image>`, *please generate a caption for the image:* "

### A.5   Visual Question Answering

**Example 1.** "*The image is* `<image>`. *Please generate an answer according to the question:* `<question>`. "

**Example 2.** "*The image is* `<image>`. *Please answer the question* `<question>` *according to the image.* "

**Example 3.** "*With the objects in the* `<image>`, `<question>`. "

## B   Experimental Settings

### B.1   Datasets

VisionLLM unifies the output formats of vision and language tasks as vocabulary generation, which enables models to be jointly trained on a wide range of tasks. In the experiments, we investigate the general modeling capacities of VisionLLM on five vision-centric tasks, including object detection, instance segmentation, visual grounding, image captioning, and visual question answering.

For object detection and instance segmentation, COCO2017 [13] is used for training and evaluation. For visual grounding, we combine the annotations of RefCOCO [27], RefCOCO+ [27] and RefCOCOg [17] for training, resulting in over 120k referred objects in total. And our models are evaluated on the validation set of RefCOCO. For image captioning and visual question answering, we adopt COCO Caption [4] and LLaVA-Instruct-150K [14] as the training source. We evaluate

the image captioning performance on the COCO Karpathy test split following common practice [10, 23, 26]. We mainly use qualitative results to demonstrate the VQA capability of our model, as LLaVA-Instruct-150K is not compatible with the standard VQA benchmark. These tasks differ in their granularity, ranging from coarse-grained image level to fine-grained pixel level, enabling a comprehensive evaluation of the model's ability to adapt to different levels of customization through language instructions.

## B.2 Implementation Details

We implement two variants of VisionLLM with two image backbones, *i.e.*, ResNet [8] and InternImage-H [24]. ResNet-50 is initialized with ImageNet-1K [5] pre-trained weights. InternImage-H is firstly pre-trained on ImageNet-22K and then trained on the detection task with Objects365 [20]. For the language-guided image tokenizer, we adopt BERT-Base [1] as the text encoder and Deformable DETR (D-DETR) [28] to capture high-level information. We set the number of queries $M$ to 100, and the number of encoder/decoder layers to 6 for D-DETR. For the LLM, we employ Alpaca-7B [21], a LLaMA [22] model fine-tuned with instructions, and equip it with LoRA [9] for parameter-efficient fine-tuning.

The model is trained in two stages. In the first stage, we initialize the model with the pre-trained weights of D-DETR, BERT, and Alpaca-7B, and train the visual backbone and the language-guided image tokenizer, while freezing most parameters of the LLM except a few LoRA parameters. To simplify the training complexity, in this stage, we mainly focus on object detection tasks with random object categories and task descriptions. In the second stage, we freeze the visual backbone and introduce the unified supervision of multiple tasks. Unless otherwise specified, the training runs for 50 epochs on $4 \times 8$ NVIDIA A100 GPUs. AdamW [15] is used as the optimizer, with one sample per GPU. We employ the cosine annealing schedule [16] as the learning policy, with an initial learning rate of $2 \times 10^{-4}$. In addition to the experiments in the main paper, more experimental settings and ablation studies are provided in the supplementary material due to space limitations.

## C Output-Format-as-Query Decoding

The output-format-as-query decoding technique is designed to parse the user instructions into the standard output format, which is compatible with the LLM-based decoder. In this section, we introduce its details from data construction, training, and inference aspects.

### C.1 Data Construction.

Following self-instruct [25], we create various user instructions for each task to simulate human interaction. Here is an example:

**System Message.** "You are an AI assistant for translating the user instructions to the standard prompt. Please help me parse the following input. Input: {input} Output:"

**Object Detection.** "Input: The image is: <image>. Please thoroughly examine the image and detect all objects belonging to the category set 'person': <c0>, 'bicycle': <c1>, 'car': <c2>, 'motorcycle': <c3>. Output: The bounding boxes are <cls> <x1> <y1> <x2><y2> <cls> <x1> <y1> <x2> <y2> ... <cls> <x1> <y1> <x2><y2>."

**Image Caption.** "Input: The image is: <image>. Please write a short caption for this image. *Output:* The image shows that <bos>"

### C.2 Training and Inference

**Training.** After obtaining the data of user instructions, we finetune Alpaca using the next token prediction task for supervision, making it able to accomplish the output format parsing process.

**Inference.** As described in Sec. 3.4 and Figure 4, the inference process involves the following steps:

(1) We first use the fine-tuned Alpaca to parse the user instructions into standard output formats for different tasks. For instance, in the case of object detection, the output format may be "The bounding

Table A: Comparison of the time cost of different methods.

| Method | FPS | Times per Image |
|---|---|---|
| VisionLLM-R50 | 5.1 img/s | 197.4 ms |
| Pix2Seq-R50 | 4.4 img/s | 227.3 ms |
| VisionLLM-ViT-B | 4.0 img/s | 251.7 ms |
| Pix2SeqV2-ViT-B | 3.4 img/s | 294.1 ms |

boxes are <cls> <x1> <y1> <x2> <y2> <cls> <x1> <y1> <x2> <y2> ... <cls> <x1> <y1> <x2> <y2>". For image captioning, the output format could be "The image shows that <bos>".

(2) The parsed outputs are then appended to the original user instructions as suffix texts. The extended instructions are fed into the LLM-based decoder as queries.

(3) Since the output format contains special tokens, such as <cls>, <x1>, <y1>, <x2>, <y2>, and <bos>, by treating these tokens as queries, the LLM-based decoder can predict the corresponding results. This approach enables the detection task to run in parallel like the cloze task, while the captioning task remains the next token prediction.

## D   Time Cost Analysis

As shown in Table A, we compare the inference speed of Pix2Seq and VisionLLM. Specifically, we executed tests on a single A100 GPU, utilizing code and model weights from Pix2Seq's official repository. For both methods, we set the batch size as 1 and the image size as 1024x1024. We see that although VisionLLM is equipped with a large LLM-based decoder, its inference speed is faster than Pix2Seq. This shows that VisiomLLM has an acceptable inference speed thanks to the proposed output-format-as-query decoding.

## E   Differences between VisionLLM and Pix2Seq Series

Although both VisionLLM and Pix2Seq v1/v2 employ coordinate discretization for object detection tasks, they differ significantly in terms of task generality, model design, and decoding process.

**Task Generality.** VisionLLM allows users to customize vision tasks using language instructions, supporting user-tailored output formats, task targets, task descriptions, etc. In contrast, Pix2Seq v1 is a special model for object detection, and Pix2Seq v2 only supports pre-defined task switching with learnable prompt tokens, lacking the flexibility of task customization.

**Model Design.** VisionLLM consists of a series of careful designs for open-ended tasks, including (1) language instructions that align vision tasks with NLP tasks; (2) a flexible tokenizer guided by natural language instructions (Pix2Seq v2 uses unreadable embedding for task switching); and (3) an open-ended task decoder based on LLMs along with an improved decoding process.

**Decoding Process.** Pix2Seq struggles to converge in open-ended task scenarios with random user instructions (see Table A(d) in supplementary material). VisionLLM solves this problem effectively by using its output-format-as-query approach, which enables the model to work with the Hungarian matching loss and handle highly random open-ended task instructions efficiently.

To sum up, VisionLLM and Pix2Seq are distinct models. Pix2Seq is a pioneering generalist model but has limitations (see Figure 1(a)). VisionLLM explores new possibilities for end-to-end models that unify vision and language tasks in the LLM era.

## F   Loss Function

VisionLLM consists of two model components: language-guided image tokenizer and LLM-based open-task decoder. So the total loss $\mathcal{L}$ of our model can be written as:

$$\mathcal{L} = \mathcal{L}_{\text{tok}} + \mathcal{L}_{\text{dec}}, \tag{1}$$

where $\mathcal{L}_{\text{tok}}$ and $\mathcal{L}_{\text{dec}}$ denote the loss of language-guided image tokenizer and LLM-based open-task decoder, respectively. We introduce the two loss functions as follows:

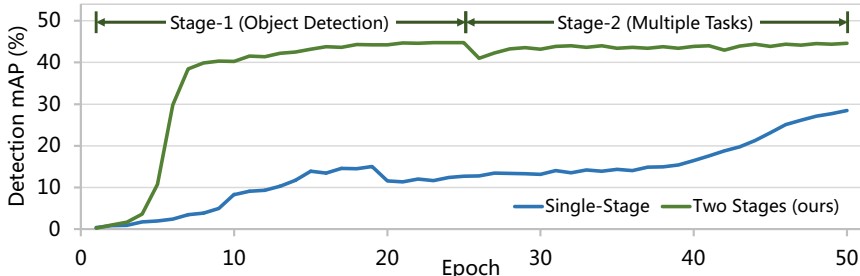

Figure A: **Comparison of two training schedules for VisionLLM.** We found that a two-stage training from easy to hard converges faster than a single-stage training.

**Language-Guided Image Tokenizer.** Different from the Q-Former [11], we use a supervision method similar to that of Deformable DETR [28], but with a different loss $\mathcal{L}_{\text{tok}}$: category-agnostic classification (focal loss [12]) and center point regression ($L_1$ loss). As explained in Sec. 3.3, our image tokenizer extracts $M$ image tokens $T = \{(e_i, l_i)\}_{i=1}^{M}$, each of which is represented by an embedding $e_i$ and a location $l_i$ (*i.e.*, absolute coordinates of the center point).

**LLM-Based Open-Ended Task Decoder.** We handle two cases in decoding processing differently. (1) For regular word prediction, we train with standard next-token supervision [22, 18, 2, 19]; (2) For unordered set prediction (*e.g.*, bounding boxes), we first output a sequence of tokens according to the output format (see the output-format-as-query paradigm in Sec. 3.4), then use bipartite matching to align the LLM-predicted outputs with the ground truths. Despite the differences, we use cross-entropy to compute the loss $\mathcal{L}_{\text{dec}}$ in a unified way for both cases.

## G  Training Schedule

As shown in Figure A, to speed up the convergence of VisionLLM, we split the training schedule of VisionLLM into two stages:

**Stage 1.** In this stage, we initialize the language-guided image tokenizer by loading the pre-trained weights of Deformable DETR [28] and BERT [6]. Additionally, Alpaca [21] is employed as the LLM-based open-ended task decoder. To align visual tokens with text tokens, we make the language-guided image tokenizer trainable while freezing most parameters of the pre-trained Alpaca, with only a few LoRA [9] parameters left tunable. We only focus on object detection in this stage to simplify the training difficulty, with random task descriptions and object categories.

**Stage 2.** The second stage builds upon the model weights obtained from the first stage. For efficiency, we freeze the visual backbone (*e.g.*, ResNet [8]) in the language-guided image tokenizer. Notably, this stage introduces the unified supervision of multiple tasks, including object detection, instance segmentation, visual grounding, image captioning, and VQA, facilitating the model to leverage the power of LLMs to understand and manipulate visual information holistically.

## H  More Ablation Studies

In this section, we provide more ablation studies and analysis of VisionLLM. Unless otherwise specified, we use ResNet-50 [8] backbone and perform the ablation experiments for object detection tasks with random task descriptions and object categories on COCO 2017 [13].

**Randomness.** In Table Ba, we examine the effect of introducing randomness during training for VisionLLM, including randomness in task descriptions, object categories, and output formats (*i.e.*, multi-task joint training). Initially, without any randomness, the model achieves a box AP of 45.2. However, as randomness is gradually applied, interesting phenomena emerge: while there is a slight decrease (45.2 → 44.6) in the AP of standard detection with the introduction of randomness, the overall benefits of enhanced task customization and open-ended capabilities outweigh this minor trade-off. Overall, introducing randomness during training in VisionLLM positively impacts its capacity for open-ended tasks and customization.

Table B: **More ablation studies for VisionLLM.**

(a) Effect of randomness.

| Randomness | AP |
|---|---|
| None | 45.2 |
| + Random Task Description | 45.1 |
| ++ Random Object Category | 44.8 |
| +++ Random Output Format (Multi-task Joint Training) | 44.6 |

(b) Effect of LoRA [9].

| LoRA | Randomness | AP |
|---|---|---|
| ✗ | ✗ | 45.2 |
| ✗ | ✓ | 1.2 |
| ✓ | ✓ | 44.8 |

(c) Effect of the number of image tokens.

| #Tokens | AP |
|---|---|
| 50 | 44.5 |
| 100 | 44.8 |
| 200 | 45.1 |
| 300 | 45.2 |

(d) Effect of Seq2Seq.

| Seq2Seq | AP |
|---|---|
| ✓ | - |
| ✗ | 44.8 |

(e) Large vocabulary object detection.

| Dataset | #Classes | AP |
|---|---|---|
| COCO | 80 | 44.8 |
| LVIS | 1203 | 18.9 |

**Low-Rank Adaptation (LoRA).** As shown in Table Bb, when randomness is not applied, the model achieves $45.2$ box AP without using LoRA [9]. However, when randomness is employed, it is observed that the model fails to converge without using LoRA. Conversely, when LoRA and randomness are used together, the model is able to converge. This indicates that LoRA plays a crucial role as a bridge between the language and visual tokens, enabling effective alignment between the two modalities and improving the convergence of the overall system.

**Number of Image Tokens.** We vary the number of image tokens from $50$ to $300$ to investigate their impact on the performance. Results are presented in Table Bc. As the number of image tokens increases, the performance continues to improve. This makes sense because a larger number of image tokens provides a more detailed description of the image content. Considering computational complexity, we adopted $100$ image tokens in our experiments.

**Robustness to Prompt Changes.** Since VisionLLM is trained with random prompts, including random task descriptions and random categories, one may ask whether there is a large performance variance across different prompts. To validate the stability of VisionLLM, we conduct experiments using eight different prompts. The first six prompts employ different task descriptions, while the last two prompts involve random category orders. In the case of random category orders, we map the categories back to the COCO standard category order for evaluation. As shown in Figure B, most evaluation results are distributed closely to $44.8$ AP. The performance differences among prompts are marginal, demonstrating that VisionLLM is robust to different prompts.

**Instruction Following Capability.** As shown in Figure C, when the prompt only contains $40$ classes, the performance for these categories remains normal, while the performance for the remaining categories is close to zero. This indicates that VisionLLM can dynamically detect objects based on the given class set <class> in instructions while disregarding the other classes that are not mentioned. This result highlights the flexibility of VisionLLM in adhering to instructions.

**Output-Format-As-Query *vs*. Seq2Seq.** In VisionLLM, we introduce the output-format-as-query framework for LLM decoder. Alternatively, we also experiment with the sequence generation method like Pix2Seq [3] for object detection with random task descriptions and object categories. However,

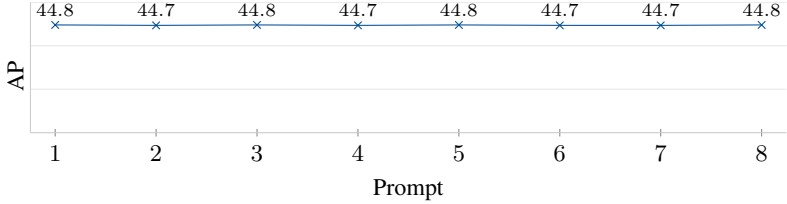

Figure B: **Evaluation results using eight different prompts.** The first six prompts use different task descriptions of object detection, while the last two prompts employ random category orders. These results show that the performance of different prompts is similar, only a $0.1$ AP gap is observed.

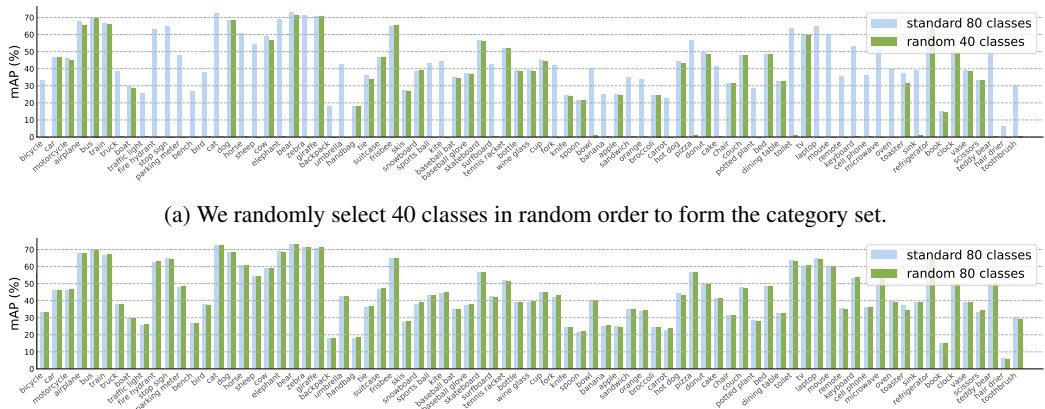

(a) We randomly select 40 classes in random order to form the category set.

(b) We randomly change the order of 80 classes to form the category set.

Figure C: **Per-category AP on COCO dataset.** We randomly select some categories to form the category set `<class>` in language instructions.

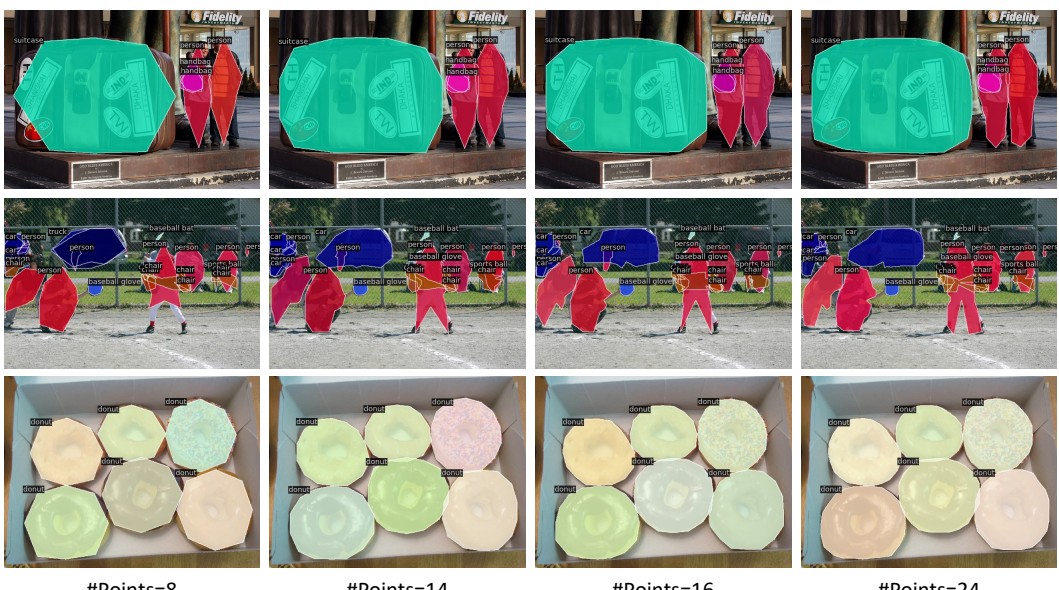

| #Points=8 | #Points=14 | #Points=16 | #Points=24 |

Figure D: **Customization of instance masks using the different number of points.** Notably, we only modify the output format mentioned in the prompt, *i.e.* the number of segmentation points. For more details, please see the example prompts provided in Sec. A.2.

we find that the loss is hard to converge in this paradigm, which indicates that the seq2seq decoding may need a more detailed design or a longer training schedule for the open-ended visual tasks, while the proposed output-format-as-query framework is more effective for open-ended tasks.

**Large-Vocabulary Object Recognition.** To validate the capacity of VisionLLM in the large-vocabulary scenario, we further conduct the experiments on the challenging dataset LVIS [7] with 1203 categories. Due to the limited number of language tokens, we randomly select 80 classes for training in each iteration. During inference, we divide the 1203 categories into 16 groups and predict the results in a sliding-window manner. As shown in Table Be, without tricks like federal loss, VisionLLM-R50 can achieve 18.9 mAP on LVIS.

**Instruction:** "*Identify the objects in the image that belong to {'person': <c0>, …, 'frisbee': <c29>, …} and draw a bounding box around each one. The output should be a list of tuples in the format (c, x1, y1, x2, y2), where 'c' represents the index of the class label starting from 0, and x1, y1, x2, y2 are the offsets of the top-left and bottom-right corners of the box relative to the center point. The coordinates should be within <range>. The image is:* <image>"

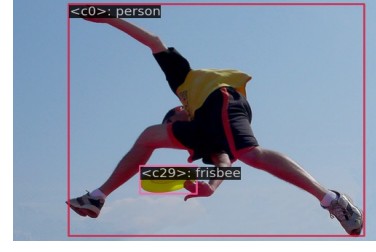

**Instruction :** "*Identify the objects in the image that belong to {'frisbee': <c0>} and draw a bounding box around each one. The output should be a list of tuples in the format (c, x1, y1, x2, y2), where 'c' represents the index of the class label starting from 0, and x1, y1, x2, y2 are the offsets of the top-left and bottom-right corners of the box relative to the center point. The coordinates should be within <range>. The image is:* <image>"

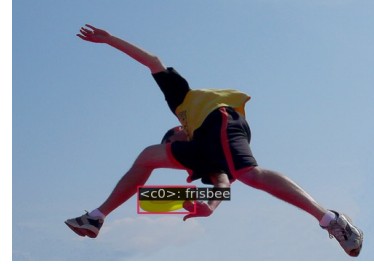

---

**Instruction:** "*Identify the objects in the image that belong to {'person': <c0>, 'bicycle': <c1>, …, 'backpack': <c24>, …, 'toothbrush': <c79>} and draw a bounding box around each one. The output should be a list of tuples in the format (c, x1, y1, x2, y2), where 'c' represents the index of the class label starting from 0, and x1, y1, x2, y2 are the offsets of the top-left and bottom-right corners of the box relative to the center point. The coordinates should be within <range>. The image is:* <image>"

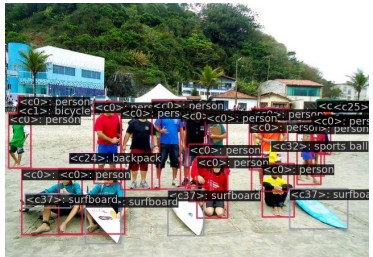

**Instruction:** "*Identify the objects in the image that belong to {'bicycle': <c0>, 'surfboard': <c1>, 'sports ball': <c2>, 'backpack': <c3>, 'the man wearing blue T-shirt': <c4>} and draw a bounding box around each one. The output should be a list of tuples in the format (c, x1, y1, x2, y2), where 'c' represents the index of the class label starting from 0, and x1, y1, x2, y2 are the offsets of the top-left and bottom-right corners of the box relative to the center point. The coordinates should be within <range>. The image is:* <image>"

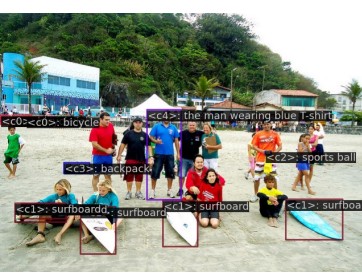

Figure E: **Customization of category set.** These results demonstrate that VisionLLM can correctly detect the specified objects according to the category set <class> provided in language instructions.

# I   Qualitative Analysis

**Customization of Segmentation Points.** In this experiment, we focus on *modifying the output format mentioned in the prompt*, specifically the number of points for instance segmentation (see Sec. A.2). The results are visualized in Figure D. Remarkably, by increasing the number of points for segmentation, we observe that the model successfully predicts more refined object masks. This validates the capability of our method to precisely customize the output format, showcasing fine-grained control over the segmentation process.

**Customization of Category Set.** We change the content of the category set <class> in language instructions and visualize the predictions in Figure E. It is observed that the model can correctly predict the object category according to the provided category set. This demonstrates that VisionLLM has a strong capacity to understand and reason over the semantic information of language instructions, which allows it for flexible category customization in open-vocabulary scenarios.

**Instruction:** "*The image is:* <image>. *Analyze the image in a* ==comprehensive== *and* ==detailed== *manner."*

**VisionLLM:** "*The image features a busy city street with a large red double-decker bus driving down the road. There are several people walking on the sidewalk, some of whom are carrying backpacks. The bus is positioned towards the left side of the image. There are several people walking along the sidewalk, with some closer to the bus and others further away. The overall atmosphere suggests a busy urban environment with people going about their daily routines."*

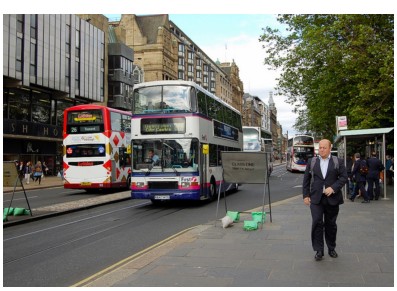

**Instruction:** "*The image is:* <image>. *Write a* ==detailed== *description of the given image."*

**VisionLLM:** "*The image features a large, open-air market filled with a variety of fruits and vegetables. There are numerous mangoes, watermelons and apples displayed in the market, with some fruits placed closer to the foreground and others further back. The market is bustling with activity as people can be seen browsing and purchasing the fresh produce. The market is likely a popular destination for those seeking a fresh and healthy diet."*

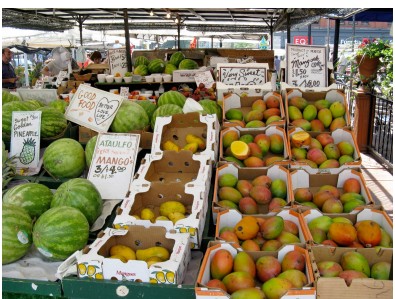

**Instruction:** "*The image is:* <image>. *Write a* ==short== *description of the given image."*

**VisionLLM:** "*The image features a lively open-air market with a variety of fruits and vegetables. People are busy browsing and buying fresh produce."*

(a) Image description with controllable text length

**Instruction:** "*The image is:* <image>. *Please generate an answer according to the question:* ==what reasons could there be for the cat squatting on the shoes?==*"*

**VisionLLM:** "*There could be several reasons. One possibility is that the cat is seeking warmth or comfort, as shoes can provide a cozy and warm surface for the cat to rest on. Another reason could be that the cat is seeking attention or affection from its owner, who is standing nearby. Cats are known to be curious creatures, and they may be seeking attention or attention from their owners."*

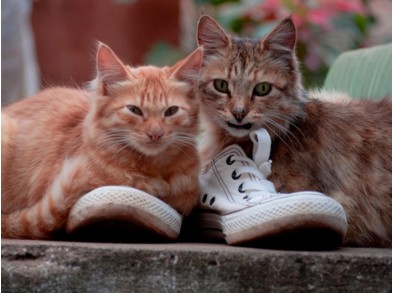

**Instruction:** "*The image is:* <image>. ==What do you think of the working environment here?==*"*

**VisionLLM:** "*The working environment in this office space appears to be open, spacious, and well-equipped. There are several desks set up for employees, providing ample workspace for each individual. The office features a modern and contemporary design, which suggests a creative and collaborative work environment. The presence of several chairs and the organization of the office, including the laptops and monitors, indicate that the space is well-maintained and conducive to productivity. Overall, the office appears to be a pleasant and efficient work environment for its employees."*

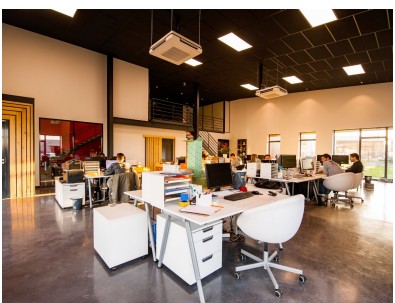

(b) Visual question answering with reasoning

Figure F: **Visualization of the image description and VQA capabilities of VisionLLM.**

**Image Description & VQA.** Benefiting from the power of LLMs, VisionLLM exhibits a strong ability in generating long descriptions for images and answering visual questions with complex reasoning. We show the examples in Figure F.