# OpenReview forum: "VisionLLM: Large Language Model is also an Open-Ended Decoder for Vision-Centric Tasks"
_NeurIPS.cc/2023/Conference — NeurIPS 2023 poster_

### Official Review · Reviewer_eUTR · 2023-06-27

**Soundness:** 3 good
**Presentation:** 2 fair
**Contribution:** 3 good
**Rating:** 7
**Confidence:** 4

**Summary:**

The paper presents a new vision-language model where tasks are specified via a language interface instead of being “hard-coded” into the architecture. The model relies on a pretrained decoder model to simply output the answer as text. E.g., for the object detection task the model would output a set of coordinates directly. The model architecture is rather complicated – a backbone encodes the input image into a multi-scale feature map. The language information is encoded by a Bert model and injected into the multi-scale feature map via cross-attent. Then a DETR model produces a set of visual “tokens” from the feature map. These tokens are then fed to the decoder which outputs the answer in natural text. The paper presents strong results on object detection, grounding and captioning.

**Strengths:**

* The paper provides very strong empirical results. Especially the object detection scores (60ap) are close to SOTA despite this being essentially zero-shot.
* The paper is relatively easy to train -- it relies on LORA and pre-trained models. This and the open-sourced code should make the method available to other researchers.
* Decoding the output in natural text is a good research direction, it is much more flexible than visual prompt tuning and predefined formats.


**Weaknesses:**


* The method is rather complicated. There are many components and exactly how they interact is not clear from the paper. As a reader, I would probably not be able to implement this from the paper, and I don’t really know what the crucial components are. I have many questions below which I hope the authors can answer to improve the presentation. E.g. the output-format-as-decoding method is not clearly described.

* There are also VL tasks (e.g. VQA) which I think should be added to the main paper. Currently only detection/segmentation, grounding and captioning are available.




**Questions:**

1. Could you provide pseudo-code for your implementations? E.g. you say that “The language features are then injected into each scale of visual features through cross-attention”. Knowing exactly how the cross-attention is implemented would be good.

2. What is output-format-as-decoding? The paper says that you “feed the tokens of structural output format as queries to the decoder”, can you explain in more details what this means?

3. The paper says “except a few LoRA parameters” – can you specify how many?

4. Do you know how much object specific knowledge is available in the pretrained DETR model? Is it possible to do an ablation here? I assume the DETR model has been pretrained on e.g. COCO detection, so it’s maybe not really a zero-shot task for the model?

5. Why is two-stage training needed? Are there ablations showing the effects of this?

6. Are the resnet parameters pretrained? The papers just says “we initialize the model with the pre-trained weights of D-DETR, BERT, and Alpaca-7B”.

---

> ### Author Rebuttal · Authors · 2023-08-10
>
> **Q1: There are many components and exactly how they interact is not clear from the paper.**
>
> **A1:** There are only two main components in the VisionLLM: language-guided image tokenizer and LLM-based decoder, each of which has specific designs for open-ended tasks. We kindly invite the reviewer to see Reviewer wdrB's Q1 for more clarification and detailed interaction among components. We will make it clearer in our revised version.
>
> **Q2: Details of `output-format-as-query` decoding.**
>
> **A2:** Thanks for your good question. Since it was also raised by other reviewers, we provide details about output-format-as-query in terms of data construction, training, and inference process in Common Questions Q1. We will make it clearer in our revised version.
>
> **Q3: There are also VL tasks (e.g. VQA) which I think should be added to the main paper.**
>
> **A3:** For VQA tasks, we qualitatively showcase the performance of VisionLLM on complicated VQA scenarios, as shown in Figure 2(d) and Figure F in the supplementary material. The reason is that conventional VQA metrics are unsuitable for measuring long and detailed answers (see L259-260), and the current GPT-based evaluation is unstable enough, see Reviewer wdrB's Q2 for more details.
>
> **Q4: The language features are then injected into each scale of visual features through cross-attention.**
>
> **A4:** Thanks for your careful review. Here we provide the pseudo-code of the cross-attention layer. In this code, we inject text features into each scale of visual features through cross-attention.
>
> ```python
> def vision_language_fusion(img_features, text_features, text_masks):
>     outs = []
>     for img_feature in img_features:
>         img_feature = rearrange(img_feature, 'b c h w -> b (h w) c')
>         img_feature = cross_attention(src=img_feature, ref=text_features, key_padding_mask=text_masks)
>         img_feature = rearrange(img_feature, 'b (h w) c -> b c h w')
>     outs.append(out)
>     return outs
> ```
>
> **Q5: the paper says "except a few LoRA parameters" – can you specify how many?**
>
> **A5:** We set the LoRA rank to 64 and use LoRA on the QKVO (Query, Key, Value, and Output) in the attention layers, resulting in approximately 0.9% of trainable parameters. We will clarify this in the revised version.
>
> **Q6: Do you know how much object specific knowledge is available in the pretrained DETR model? Is it possible to do an ablation here? I assume the DETR model has been pretrained on e.g. COCO detection, so it’s maybe not really a zero-shot task for the model?**
>
> **A6:** The determination of whether it is a zero-shot scenario depends on the training and test samples. As Deformable DETR model trained on the COCO dataset acquires knowledge specific to objects, it cannot be truly considered as achieving zero-shot performance for COCO evaluation. Nonetheless, it is important to note that VisionLLM is specifically designed for open-ended tasks. During testing, _**we employ instruction descriptions with arbitrary object categories, arbitrary task descriptions, and output formats, which are unseen during training**_. By leveraging the vast world knowledge embedded in a pre-trained LLM, we have observed exceptional performance of our model in comprehending and effectively handling previously unseen categories, such as "red gamepad" (see Figure 1(a)), which are not included in the COCO categories.
>
> **Q7: Why is two-stage training needed? Are there ablations showing the effects of this?**
>
> **A7:** Thanks for your careful review. As depicted in Figure A in the supplementary material, we adopted a two-stage training approach to expedite the convergence of VisionLLM. Our experiments showed that the two-stage training, starting from easy to hard tasks, resulted in faster convergence than a single-stage training approach.
>
> **Q8: Are the ResNet parameters pretrained? The paper just says “we initialize the model with the pre-trained weights of D-DETR, BERT, and Alpaca-7B”.**
>
> **A8:** Yes, the ResNet parameters are loaded from the pre-trained weights of Deformable DETR.

---

> > ### Comment · Reviewer_eUTR · 2023-08-12
> > **reply**
> >
> > Thanks for your detailed reply and clarifications. I will retain my score and increase my confidence.

---

> > > ### Author Response · Authors · 2023-08-13
> > >
> > > Thank you for your recognition. Your feedback is highly valuable to us. We will carefully consider your suggestions and continuously improve our work.

---

### Official Review · Reviewer_87vj · 2023-06-27

**Soundness:** 2 fair
**Presentation:** 3 good
**Contribution:** 2 fair
**Rating:** 5
**Confidence:** 5

**Summary:**

This work presents a LLM-based framework VisionLLM for vision-centric tasks. VisionLLM treats images as a foreign language and aligns vision-centric tasks with language tasks using language instructions. Extensive experiments show that VisionLLM deliver comparable performance  with task-specific models over different vision-centric tasks.

**Strengths:**

The authors demonstrate the feasibility of using large language models as visual decoders in which vision information is treated as a foreign language. With this idea as the key mind, authors introduce  a series of methods to aligned vision tasks in a matched format with LLMs, including designing language instruction, design the decoding process and adding additional vocabulary tokens. Conducted experiments demonstrate the feasibility of such pix2seq modelling can be scaled up as a generalist model. Glad to see that some possible limitations of this technical route are discussed for analyzing the gaps in experimental results.

**Weaknesses:**

This work follows the idea of pix2seq to build a generalist model and the authors believe that it is natural to do so, without providing a sufficient explanation of the motivation behind it. Despite the popularity of LLMs recently, we should not blindly apply them to other modalities without in-depth consideration except we can clarify the benefits and demonstrate them accordingly. Authors claim that the reasoning abilitis and world knowledge of LLMs help vision tasks, but only numerical results can be viewed as kind of evidence for this claim. It's not clear and convincing for me from two aspects:
- Whether the reasoning abilities and world knowledge of LLMs truly help? and how?
- How about scaling up task-specific vision models when compared to VisionLLM? In other words, how can you demonstrate the performance superiority comes from using LLMs as vision decoders, instead of model scaling up?

There are a lot of teachnical designs introduced to model vision-centric tasks in a language-matched format. The reasonableness of some of these are hard to be guaranteed, which is subject to further discussion. One of the strangest designs in VisionLLM is the addition of extra tokens in the vocabulary to represent the position values and categories. This approach appears to go against the initial purpose of leveraging LLMs' world knowledge to gain an advantage in visual tasks. Besides,these additionally added tokens may cause ambiguities with those original ones in the vocabulary. I am looking forward to authors' understandings or/and some necessary evidences towards these.

The effevetiveness of VisionLLM is sorely evaluated on ResNet-50 and Intern-H backbones. The general applicability is subject to further evaluation.

**Questions:**

Some questions have been listed in the weakness part. Other than that, more questions about the details are as follows：
1. How do you handle the <image> placeholders in the actual input of VisionLLM? Do you replace them with actual image tokens in the latent space?
2. For instance segmentation, the value of N varies for different objtects. How do you detetermine this value?
3. How are "tasks defined by instructions" parsed into formatted queries in "output-format-as-query" decoding? By pre-defined templates for different task categories?

**Limitations:**

This paper tries to catch the takeoff of LLMs by building a LLM-based generalist model for vision tasks. But it cannot provide clear and convincing motivation statement, and cannot dive into the rationale of the proposed technical designs. There are also some unclear method statements as listed in my questions.

Overall, the idea behind is straightforward. But I think the entire community should be even more careful for these straightforward ideas, and think more in-depthly about whether they are truly reasonable and whether they can deliver real contributions.

---

> ### Author Rebuttal · Authors · 2023-08-10
>
> **Q1: This work follows the idea of pix2seq to build a generlist (generalist) model and the authors believe that it is natural to do so, without providing a sufficient explanation of the motivation behind it.**
>
> **A1:** We argue that _**VisionLLM is not the scaling up of Pix2Seq**_. Although both models incorporate coordinate discretization for object detection, they differ significantly in task generality, model design, and decoding process, as explained in Common Questions Q2. Moreover, directly scaling up Pix2Seq cannot achieve an open-ended task model. Such a model would not converge, as we discussed in L130-135 of the supplementary material and Table A(d).
>
> For the motivation behind this work, please note that this work focuses on open-ended vision tasks that can be customized by users according to their needs. We have summarized the limitations of the existing paradigms for open-ended tasks in Figure 1 and the first two paragraphs of the Introduction, which inspire us and shape the main objective of our work. To achieve open-ended tasks, we tackle the challenge of aligning vision tasks and LLMs from various aspects including: instruction design, model design, and training. We have explained the motivation for each design choice in the first paragraph of each subsection of the method section.
>
> **Q2: Whether the reasonding (reasoning) abilities and world knowledge of LLMs truely (truly) help? and how?**
>
> **A2:** LLM plays a crucial role in our open-ended task framework, for the following reasons:
>
> (1) LLM can parse instructions, which is a key feature of our system. As we explained in Reviewer TifL's Q1, our system cannot converge without instruct-tuned LLM. Furthermore, instruction parsing enables models to comprehend the target object mapping and the output format of the perception tasks. LLM is the only model that possesses this feature, as it is pre-trained on a large corpus of user instruction and code data.
>
> (2) LLM also facilitates image description with controllable text length and visual question answering with complex reasoning, as illustrated in Figure 2(c)(d), and Figure F in supplementary material. These tasks require the capabilities of instruct-following and relation reasoning among objects. These capabilities are not learned from visual data and models, but from LLMs pre-trained on web-scale NLP data, as demonstrated by [1].
>
> **Q3: How about scaling up task-specific vision models when compared to VisionLLM? In other words, how can you demonstrate the performance superiority (that) comes from using LLMs as vision decoders, instead of model scalling (scaling) up?**
>
> **A3:** This is a common misconception. Model scaling up could not lead to open-ended task capability. Large-scale vision models like ViT-22B, Swin-G, and InternImage-G are still limited by tasks in specific formats. They cannot handle vision tasks with different or unknown formats. _**So even if we scale up task-specific vision models, they are still not comparable to our model in terms of open-ended tasks.**_ On the contrary, LLM pre-trained on web-scale corpus is proved to effectively understand the user instructions and provide reasonable answers, which is important for VisionLLM. The benefits of LLM for our model can refer to Reviewer 87vj's Q2.
>
> **Q4: One of the strangest designs in VisionLLM is the addition of extra tokens in the vocabulary to represent the position values and categories. Besides, these additionally added tokens may cause ambiguities with those original ones in the vocabulary.**
>
> **A4:** _**Extra tokens are a common way to extend the capabilities of an LLM**_, especially when supporting a new language and task [2]. To support vision tasks, the original vocabulary of LLM is not enough (see L210-213), it is necessary to increase the token size and ensure alignment during training. These tokens will be aligned with the LLM by instruction tuning. _**They will not conflict with the original tokens**_, because in our constructed training data, the newly added tokens are only used for vision tasks without any overlapped semantics with the original tokens.
>
> **Q5: The effectiveness of VisionLLM is sorely evaluated on ResNet-50 and Intern-H backbones. The general applicability is subject to further evaluation.**
>
> **A5:** We evaluate VisionLLM on two representative backbones of different scales. ResNet-50 is the most representative common-scale backbone, while InternImage-H is a large-scale backbone with state-of-the-art performance. We provide additional experiments using ViT-B as a vision encoder, through which we reach similar conclusions as ResNet-50 and InternImage. See Common Questions Q3 for more details.
>
> **Q6: How do you handle the <image> placeholders in the actual input of VisionLLM? Do you replace them with actual image tokens in the latent space?**
>
> A6: Yes, for `<image>`, we replace it with image tokens in the latent space.
>
> **Q7: For instance segmentation, the value of N varies for different objects. How do you determine this value?**
>
> **A7:** During training, the number of points varies randomly with the language instruction. During inference, the number of points N is specified by the user in his/her instructions.
>
> **Q8: How are "tasks defined by instructions" parsed into formated (formatted) queries in "output-format-as-query" decoding? By pre-defined templates for different task categories?**
>
> **A8:** See details of the output-format-as-query decoding in Common Questions Q1. We introduce the data construction, training, and inference details of the decoding process.
>
> [1] Ouyang, Long, et al. "Training language models to follow instructions with human feedback." Advances in Neural Information Processing Systems 35 (2022): 27730-27744.
>
> [2] Schick, Timo, et al. "Toolformer: Language models can teach themselves to use tools." arXiv preprint arXiv:2302.04761 (2023).

---

> > ### Comment · Reviewer_87vj · 2023-08-14
> > **Reply to Rebuttal**
> >
> > Thank you for your efforts in providing the rebuttal. However, some of my questions have not been fully understood, and the current rebuttal has not convinced me and addressed my concerns. I look forward to a more in-depth discussion.
> >
> > In my review comments, I did **NOT** suggest that "***VisionLLM is not the scaling up of Pix2Seq.***" What I would like to discuss is whether the modeling of VisionLLM is reasonable enough. In VisionLLM, visual information (e.g., coordinates) is decoded by a LLM using additionally added tokens. Such modelling raises a series of questions that we really need to take seriously:
> >
> > **How does the using of LLM impact the vision tasks themselves?** It is a common sense that LLM can understand instructions and enable the versatility over different tasks. What I am concerned is the impact of LLM when applied to visual tasks. Is its impacts positive or negative for vision tasks? And WHY?
> > As shown in your provided experiment results in this rebuttal, the proposed VisionLLM-ViT-B has lower AP on Instance Seg. and lower BLEU-4 on Captioning compared to Pix2Seq v2 (ViT-B), when they use the same vision encoder. Is the performance drops of VisionLLM caused by the different model it use? How about using a vision decoder (or a vision decoding head) instead of a language decoder?
> >
> > **How does the using of additionally added tokens impact the effectiveness?** "Extra tokens are a common way" has never been a responsible answer for "Is this modeling correct (although it is often used)? ". After reading authors' responses, I am still confused about:
> > - Does adopting additionally added tokens run counter to the open-ended purpose? For instance, the added classification tokens correspond to classification semantics in a deterministic way as stated in Line224, and their number is limited. What if the target category falls ouside of these added classification tokens? Is this a close-set limitation against open-ended?
> > - Why don't these additionally added markers cause ambiguity with the original markers in the vocabulary? For example, tokens for numbers (0~9) simultaneously exist in both original tokens and additionally added tokens. How does the model handle them?
> > - Why use auto-regressive decoding for the additionally added tokens? When we expand the vocabulary of LLM with additional tokens, we need to decode these tokens auto-regressively as for other tokens in the vocabulary. Compared to directly regress coordinates or perform classification over these tokens, is this auto-regressive decoding for these tokens a correct manner?
> >
> > I hope the authors can deeply consider these important questions and provide responsible answers. Following commonly used methods may deepen the misguidance of the research community sometimes.

---

> > > ### Author Response · Authors · 2023-08-16
> > >
> > > We appreciate your time and effort in reviewing. We are open to engaging in a more in-depth discussion regarding this work.
> > >
> > > **Q9: The statement of "VisionLLM is not the scaling up of Pix2Seq".**
> > >
> > > **A9:** In your comments, it was mentioned that "Conducted experiments demonstrate the feasibility of _**scaling up**_ such pix2seq modeling as a generalist model", and also raised the question "How about _**scaling up**_ task-specific vision models when compared to VisionLLM?". Therefore, the statement _**"VisionLLM is not the scaling up of Pix2Seq"**_ has been emphasized to clarify the contribution of this work and avoid potential misunderstanding.
> > >
> > > **Q10: How does the using of LLM impact the vision tasks themselves? It is a commen (typo: common) sense that LLM can understand instructions and enable the versatility over different tasks. What I am concerned is the impact of LLM when applied to visual tasks. Is its impacts positive or negative for vision tasks? And WHY?**
> > >
> > > **A10:** _**LLM plays a crucial role in this work due to its parsing and instruction-following capabilities. These capabilities serve as the foundation for defining and understanding open-ended descriptions of vision tasks.**_ We have explained this _**in Q2**_, and provided additional evidence _**in the follow-up question 1 of Reviewer TifL**_. Furthermore, we also explained _**in Q3**_ that simply scaling up the decoding head without web-scale corpus pre-training cannot achieve the same capability as LLM. Therefore, LLM is a positive and crucial component for open-ended vision tasks from this perspective.
> > >
> > > Based on this point, we introduce VisionLLM, a viable framework with a series of tailored designs that align vision tasks with LLMs. As we mentioned _**in follow-up question 2 of Reviewer TifL**_, due to the need to unify various vision tasks, the framework makes compromises that may affect its performance, especially in segmentation tasks. Additionally, Pix2Seq v2 has 128 polygon points, which is 4 times more than our model. It also uses ensemble and crop-then-segment techniques to enhance the segmentation results, but these techniques are independent of the open-ended task and are not considered in this work.
> > >
> > > Regarding image captioning, as we explained in _**Q2 of Reviewer kyDb**_, the linguistic capabilities of our model are aligned with LLM, resulting in longer and more detailed responses. If we discard the LLaVA-Instruct-150K dataset and train on COCO Captions, our model could achieve better performance, as demonstrated in the table below:
> > >
> > > | Model            | BLEU-4 |
> > > | ---------------- | ------ |
> > > | Pix2Seqv2-ViT-B  | 34.9   |
> > > | VisionLLM-R50    | 31.0   |
> > > | VisionLLM-R50*   | 33.0   |
> > > | VisionLLM-ViT-B  | 31.5   |
> > > | VisionLLM-ViT-B* | 35.6   |
> > >
> > > \* indicates discarding the LLaVA-Instruct-150K dataset and training on COCO Captions
> > >
> > > However, there is an inconsistency between the standard metric (favoring shorter text) and the user experience. If we prioritize alignment with the standard captioning benchmark, it may sacrifice user-friendliness and the overall versatility of the model. In contrast, this work aims to tackle user-defined visual tasks in a flexible manner and provide a practical framework that addresses open-ended tasks effectively. So unlike previous models (e.g., Pix2Seq v2) that pursue superior performance on _**pre-defined vision tasks**_, our approach prioritizes _**open-ended tasks**_ to meet the diverse needs of users.

---

> > > > ### Author Response · Authors · 2023-08-16
> > > >
> > > > **Q11: How does the using of addtionally (typo: additionally) added tokens impact the effectiveness? "Extra tokens are a common way" has never been a responsible answer for "Is this modeling correct (although it is ofen (typo: often) used)? "**
> > > >
> > > > **A11:** "Extra tokens are a common way" means it is a widely accepted and proven effective approach, which is supported by numerous peer-reviewed studies in the community [1, 2, 3]. To our best knowledge, there is no evidence indicating that using extra tokens is a suboptimal design for new tasks. _**It is unreasonable to neglect commonly used settings and introduce new, unverified ones.**_
> > > >
> > > >
> > > > **Q11-1: Does adopting addtionally (additionally) added tokens run counter to the open-ended purpose? For instance, the added classification tokens correspond to classification semantics in a deterministic way as stated in Line224, and their number is limited.**
> > > >
> > > > **A11-1**: No, _**the addition of tokens does not run counter to the open-ended task purpose**_. There might be a misunderstanding. As explained in L222-226, the mapping between classification tokens and semantics is fully flexible and can be defined dynamically by the user through their instructions. In other words,  the class set of our model is open. It is possible to use a question sentence or a detailed description as a class (see Figure 1(a) and Figure I in the attached PDF file for rebuttal). Although there is a limitation on the number of classification tokens, it is possible to handle an infinite number of classes through sliding-window inference. As demonstrated _**in L136-141 in supplementary material**_, we divide the 1203 categories into 16 groups and predict the results in a sliding-window manner.
> > > >
> > > > **Q11-2: Why don't these additionally added markers cause ambiguity with the original markers in the vocabulary? For example, tokens for numbers (0~9) simultaneously exist in both original tokens and (typo: addtionally) additionally added tokens. How dose (typo: does) the model handle them?**
> > > >
> > > > **A11-2:** _**No ambiguity will be included.**_ We have avoided this problem when constructing the ground truth data. Only when the query token is a token like <cls>, <x i>, and <y i>, additional tokens such as <p i> and <c i> will be output. Other common words (including numbers (0~9)) will not activate them.
> > > >
> > > > **Q11-3: Why use auto-regressive decoding for the addtionally (typo: additionally) added tokens? When we expand the vocabulary of LLM with additional tokens, we need to decode these tokens auto-regressively as for other tokens in the vocabulary. Compared to directly regress coordinates or perform classification over these tokens, is this auto-regressive decoding for these tokens a correct manner?**
> > > >
> > > > **A11-3:** There seems to be a misunderstanding. For example, in perception tasks, we utilized the "output-query-as-format" approach to avoid auto-regressive decoding when predicting coordinates and classes. And the exta tokens such as <p i> and <c i> are only output when the query token is <cls>, <x i>, or <y i>. This approach, similar to the format of cloze test, allows us to predict coordinates and classes in parallel, akin to the DETR model. Please refer to _**Common Questions Q1**_ for additional details.
> > > >
> > > >
> > > > &nbsp;
> > > >
> > > > Thank you for your review. Your questions and feedback have provided us with an opportunity to reflect and improve.  If you have any further questions or suggestions, we would be more than willing to continue the in-depth discussion.
> > > >
> > > > &nbsp;
> > > >
> > > > [1] Tai, Wen, et al. "exBERT: Extending pre-trained models with domain-specific vocabulary under constrained training resources." Findings of the Association for Computational Linguistics: EMNLP 2020.
> > > >
> > > > [2] Koh, Jing Yu, Ruslan Salakhutdinov, and Daniel Fried. "Grounding Language Models to Images for Multimodal Inputs and Outputs." ICML 2023.
> > > >
> > > > [3] Hong, Jimin, et al. "AVocaDo: Strategy for Adapting Vocabulary to Downstream Domain." EMNLP 2021.

---

> > > > > ### Comment · Reviewer_87vj · 2023-08-17
> > > > > **Further Reply to Rebuttal**
> > > > >
> > > > > Thanks for your efforts on rebuttal. Sorry for the typos during the quick typing. My concerns has been mostly addressed. And I will increase the final score.
> > > > >
> > > > > Besides, a further question regarding Q11-2 is: As you explain, when the query token is a token like <cls>, <x i>, and <y i>, additional tokens such as <p i> and <c i> will be output. In the original vocabulary, there are also some tokens sharing the same or similar semantics with your additional tokens. Will these tokens remain activated?

---

> > > > > > ### Author Response · Authors · 2023-08-18
> > > > > >
> > > > > > Thank you for your appreciation of our work.
> > > > > >
> > > > > > **Q12: A further question regarding Q11-2 is: As you explain, when the query token is a token like <cls>, <x i>, and <y i>, additional tokens such as <p i> and <c i> will be output. In the original vocabulary, there are also some tokens sharing the same or similar semantics with your additional tokens. Will these tokens remain activated?**
> > > > > >
> > > > > > **A12:** In the original vocabulary, there are some tokens that share semantics with the additional tokens, such as "class", "x", "y", and numbers. These tokens are common words that will be predicted using a next-token-prediction approach in tasks such as question answering and captioning. For example,
> > > > > > ```
> > > > > > Case 1: "Human: What is the class of the object? Assistant: The class of the object is a cat."
> > > > > >
> > > > > > Case 2: "Human: x=1, y=2, x+y=? Assistant: x+y=3."
> > > > > >
> > > > > > Case 3: "Human: How many cats are in the image? Assistant: 10."
> > > > > > ```

---

> > > > > > > ### Comment · Reviewer_87vj · 2023-08-21
> > > > > > > **Further Questions**
> > > > > > >
> > > > > > > Will these tokens (refering to the tokens in the original vocabulary) and the additional tokens be activated simultaneously?

---

> > > > > > > > ### Author Response · Authors · 2023-08-21
> > > > > > > >
> > > > > > > > **Q13: Will these tokens (referring to the tokens in the original vocabulary) and the additional tokens be activated simultaneously?**
> > > > > > > >
> > > > > > > > **A13:** No, as responded in Q11-2, the additional tokens such as <p i> and <c i> are only activated when specific query tokens like <cls>, <x i>, and <y i> are encountered. Despite these tokens (the additional tokens and the original tokens with similar semantics) belong to the same vocabulary, the additional tokens are instruction-tuned to respond to specific queries. To this end, we established a clear link between additional tokens and specific queries and distinguished them from the original tokens when creating the instruction-tuning data. This ensures that they are not activated at the same time.

---

### Official Review · Reviewer_TifL · 2023-06-27

**Soundness:** 3 good
**Presentation:** 3 good
**Contribution:** 3 good
**Rating:** 6
**Confidence:** 4

**Summary:**

This paper introduces VisionLLM, an instruction-following agent that can perform various vision-only (classification/detection/segmentation) and vision-language (captioning/VQA) tasks. The proposed model connects a pre-trained visual backbone with a language decoder Alpaca with a language-aware image-tokenizer. To unify vision-only and vision-language tasks, VisionLLM adopts different language instruction formats. Furthermore, it proposes an “output-format-as-query” framework for efficient parallel decoding for vision-centric tasks.

**Strengths:**

The paper presents an impressive effort in developing a large decoder for vision-only and vision-language tasks, using state-of-the-art multimodal foundational models while treating images as foreign language. The technical details are comprehensive and sound. Extensive experiments demonstrate the effectiveness of the proposed system compared to state of the art Pix2Seq approaches. Ablation studies show insights on how the system perform in single-/multi-task scenarios and different image tokenizing schemes.

**Weaknesses:**

Even though the proposed system is technically sound, it is still quite complicated. It is unclear what its major advantages are compared to previous Pix2Seq methods and task-specific models.

I have some doubts about the system design:
- Is the pre-trained instruction-following LLM (Alpaca) crucial in your system design? Can the system perform as well with a (non-instruction-following) LLaMA? Do more advanced instruction-following agents bring advantages over naive LLMs such as T5?
- One of the major disadvantages of using a large-scale pre-trained LLM (Alpaca) is the increased training and inference costs. Could you compare the efficiency of the proposed system to prior task-specific models/generalist models that do not use such large-scale pre-trained decoder?
- Why not adopt a ViT-B visual encoder to facilitate comparison to other generalist models such as Uni-Perceiver and Pix2Seq (as in Table 1)?

The paper is overall well-written. I found one typo:
L166: placeholdersok?

**Questions:**

I have some questions about technical details:
- For detection/segmentation tasks, I am confused how the proposed “output-format-as-query” approach (L227-239) works with variable numbers of objects per image. How many “<cls> <x1> <y1> <x2> <y2>” did you send to decoder during training and inference?
- How does the “output-format-as-query” approach avoid parallel decoding for image captioning (L232)? As for as I understand, the decoder still adopts casual attention masking and therefore it seems token-by-token generation is necessary.
- The proposed system supports customization of number of points for segmentation tasks — does it require specific training paradigms, i.e., balancing the number of masks with different number of points?
- Does your system generalize to unseen instructions during test time?
- What is the sampling procedure for open-ended tasks? For example, do you use top-k/nucleus sampling?

**Limitations:**

No. Please include discussions on the use of foundation models and the potential biases your system might inherit from these pre-trained models.

---

> ### Author Rebuttal · Authors · 2023-08-10
>
> **Q1: Is pre-trained instruction-following LLM (Alpaca) crucial in your system design?**
>
> **A1:** The instruction-following LLM is important for the convergence of VisionLLM. We have observed that Alpaca converges more easily than LLaMa. But the LLM is not limited to Alpaca, Flan-T5 (an instruction-following version model of T5) is also a good alternative.
>
>
> **Q2: Time cost analysis.**
>
> **A2:** Thanks for your suggestion. As shown in the following table, we compare the inference speed of Pix2Seq and VisionLLM. Specifically, we executed tests on a single A100 GPU, utilizing code and model weights from Pix2Seq's official repository.  For both methods, we set the batch size as 1 and the image size as 1024x1024.
>
> As can be seen from the table, although VisionLLM is equipped with a large LLM-based decoder, its inference speed is faster than Pix2Seq. This shows that VisiomLLM has an acceptable inference speed thanks to the proposed output-format-as-query decoding. We will add time cost analysis in our revised version.
>
> | Method          | FPS       | Times per Image |
> | --------------- | --------- | --------------- |
> | VisionLLM-R50   | 5.1 img/s | 197.4 ms        |
> | Pix2Seq-R50     | 4.4 img/s | 227.3 ms        |
> | VisionLLM-ViT-B | 4.0 img/s | 251.7 ms        |
> | Pix2SeqV2-ViT-B | 3.4 img/s | 294.1 ms        |
>
> **Q3: Why not adopt a ViT-B visual encoder to facilitate comparison to other generalist models such as Uni-Perceiver and Pix2Seq (as in Table 1)?**
>
> **A3:** We evaluate VisionLLM on two representative backbones of different scales. ResNet-50 is the most representative common-scale backbone, while InternImage-H is a large-scale backbone with state-of-the-art performance. We provide additional experiments using ViT-B as a vision encoder, through which we reach similar conclusions as ResNet50 and InternImage. See Common Questions Q3 for more details.
>
> **Q4: How many “<cls> <x1> <y1> <x2> <y2>” did you send to the decoder during training and inference?**
>
> **A4:** During both the training and inference phases, we input 100 sets of "<cls><x1><y1><x2><y2>" to the decoder, generating 100 object predictions. Those predictions with higher confidence scores will be retained, adhering to a common practice of the object detection task. We will make it clearer in our revised version.
>
> **Q5: How does the “output-format-as-query” approach avoid parallel decoding for image captioning (L232)?**
>
> **A5:** Sorry for the misunderstanding. We uniform the output format for various types of tasks in the form of natural language tokens. However, different types of tasks use different output formats, as illustrated in Figure 4 of the main paper.
>
> For perception tasks, we use "<cls><x1><y1> ..." as the output format, employing the “output-format-as-query” approach for parallel decoding.
>
> For understanding tasks, such as image captioning and VQA, we use "<bos>" as the output format, following a token-by-token generation process. We will provide a more explicit explanation of this aspect in the revised version.
>
> **Q6: Does it require specific training paradigms?**
>
> **A6:** No, VisionLLM does not require specific training paradigms. Like those of LLMs, the task instructions are randomly changed in terms of task type, task target, and output format (including the number of points).
>
> **Q7: Does your system generalize to unseen instructions during test time?**
>
> **A7:** Of course, our system could generalize to unseen instructions during test time. We incorporate randomized task descriptions, diverse task output formats, and randomized object categories during training. As a result, the system exhibits robustness in the face of changes in instructions.
>
> **Q8: What is the sampling procedure for open-ended tasks?**
>
> **A8:** We employed top-1 sampling in our approach. Utilizing more intricate sampling techniques, such as top-k sampling, could potentially lead to improved performance.

---

> > ### Comment · Reviewer_TifL · 2023-08-11
> > **Follow-up questions**
> >
> > Thanks the authors for the comprehensive rebuttal. I have a few follow-up questions:
> >
> > **1. *"Alpaca converges easily"* does not really convince me on its importance in the system design. Are there any references on *instruction-following* LLMs are easier to optimize and converge?**
> >
> > **2. Uni-Perceiver-V2 / Pix2Seq v2 are much stronger at segmentation tasks (when using the same ViT-B backbone). Why?**
> >
> > **3. Do you have any evidence on *generalizing to unseen test instruction*? To what extent does it work and when does it fail?**

---

> > > ### Author Response · Authors · 2023-08-13
> > >
> > > **Q1. "Alpaca converges easily" does not really convince me on its importance in the system design. Are there any references on instruction-following LLMs are easier to optimize and converge?**
> > >
> > > **A1:** We would like to discuss this issue from two perspectives as follows:
> > >
> > > (1) _**The LLM with instruction-following and parsing capabilities is able to effectively interpret the vision task instructions, which helps to reduce the loss during the early stage of model training.**_ To preserve the language ability of the LLM itself,  we freeze the weights LLM during training. If we use LLMs without instruction-following, the frozen LLM struggles to learn this ability. Therefore, we directly utilize the instruction-following LLMs. The example below demonstrates that Alpaca (instruction-following LLaMA) can parse category mapping relationships and the required output format for the task, while LLaMA cannot achieve this.
> > >
> > > ```
> > > System message:
> > > "Below is an instruction that describes a task. Write a response that appropriately completes the request.
> > > Input: {input}
> > > Output:"
> > >
> > > Input:
> > > "class set: {'person': <c0>, 'car': <c1>, 'table': <c2>, 'cat': <c3>, 'television': <c4>, 'a man in a black hat': <c5>}. What is the class index associated with the class 'a man in a black hat' in the given class set?"
> > >
> > > # Alpaca output
> > > Output:
> > > "The class index associated with the class 'a man in a black hat' in the given class set is 5."
> > >
> > > # LLaMA output
> > > "Answer: 0 Input: Given an array of strings, remove the last element of the array. The returned array should be a new array with the last element removed."
> > > ```
> > >
> > > (2) For _**the detailed loss curves**_ (since we cannot include figures at this time, we provide the loss values every 2000 iterations), the model trained using Alpaca reduces demonstrates faster loss reduction, particularly in the early stages. We do not rule out the possibility that LLaMA can converge if it is unfrozen and allowed to train with more epochs, but at least under our current experimental setting (frozen LLMs, 50 epochs), Alpaca is better at following instructions than LLaMA.
> > > ```
> > > # LLaMA loss
> > > 6.08, 4.61, 4.46, 4.41, 4.33, 4.22, 4.15, 4.10, 4.04, 4.01, 3.95, 3.87, 3.85, 3.98, 4.36, 4.63
> > >
> > > # Alpaca loss
> > > 5.26, 4.56, 3.97, 2.85, 2.77, 2.69, 2.60, 2.62, 2.54, 2.56, 2.51, 2.50, 2.50, 2.49, 2.47, 2.45
> > > ```
> > > We will make it clearer in our revised version.
> > >
> > > **Q2: Uni-Perceiver-V2 / Pix2Seq v2 are much stronger at segmentation tasks (when using the same ViT-B backbone). Why?**
> > >
> > > **A2:** Firstly, _**VisionLLM is distinct from models for pre-defined tasks (e.g., Uni-Perceiver-V2, Pix2Seq v2), as it has the capability to handle open-ended tasks customized by users, providing greater flexibility and versatility.**_ To be able to unify and flexibly customize tasks, we have made some design and trade-offs in terms of task formulation, model selection, and training methods, which may result in some performance losses, particularly on segmentation tasks.
> > >
> > > Compared to Uni-Perceiver-V2, our model utilizes polygons with discrete coordinates to represent instance masks, ensuring a uniform task output. _**This results in two levels of performance loss**_: (1) the conversion from mask to polygon representation results in performance degradation, and (2) the conversion of polygon coordinates to integers also incurs performance loss (see L347-L353).
> > >
> > > While Pix2Seq v2 also employs polygons to represent masks, they use _**128 polygon points**_ (at least 4 times more than our model) to enhance segmentation performance. Additionally, they utilize ensemble methods to _**merge the results of 8 inference runs**_ and _**adopt a crop-then-segment approach**_ to further improve segmentation performance. In contrast, VisionLLM requires an LLM-based decoder  to support general user instruction parsing. This restricts us from using more than 32 polygon points on existing hardware, and we have not implemented generalist-model-agnostic techniques (such as ensemble and crop-then-segment) to enhance our segmentation results.
> > > We will clarify this in our revised version.

---

> > > ### Author Response · Authors · 2023-08-13
> > >
> > > **Q3. Do you have any evidence on generalizing to unseen test instruction? To what extent does it work and when does it fail?**
> > >
> > > **A3:** There are four scenarios:
> > >
> > > (1) **Customized Detection Target:** Our model is trained on the COCO dataset, but it is not limited to detecting only the categories present in the dataset. The categories our model can detect can be question sentences or descriptions in natural language. This flexibility is demonstrated in Figure 2(a) and Figure I in the attached PDF file for rebuttal.
> > >
> > > (2) **Task Description Flexibility:** Our model supports user input in natural language at the task description level. It's important to note that even for the same task, the descriptions can vary. For example, the grounding task can be described in multiple ways, such as:
> > > ```
> > > # Long instruction
> > > Please identify all objects belonging to the category set {<expression>: <cls0>}. For each detected object, specify its location within the range <range> by determining the offsets of top-left and bottom-right corners relative to the center point. To indicate the object's class and location, provide the output in the format (c, x1, y1, x2, y2), where 'c' represents the class index starting from 0, and (x1, y1, x2, y2) correspond to the offsets of the bounding box corners. The image is: <image>
> > >
> > > # Short instruction
> > > Please locate the object mentioned in the category set {<expression>: <cls0>}. The image is: <image>
> > > ```
> > >
> > > These descriptions can have different lengths and sentence structures. We have validated the stability of our model with different prompts in Figure B in the supplementary material, showcasing its ability to generalize to random and unseen descriptions.
> > >
> > > (3) **Customized Output Formats:** Our method allows for customized output formats, even in object detection tasks. For instance, we can have the format as (c, x1, y1, x2, y2) or (x1, y1, x2, y2, c). We can also control the number and meaning of each point. For example, in Figure 1(a), if we modify the prompt to (x1, y2, x2, y1, c) (outputting the bottom left and top right points), the output can be as follows:
> > >
> > > ```
> > > "The bounding boxes are [(226.4, 347.4, 363.1, 229.8, <c0>), (441.1, 269.9, 538.6, 183.5, <c1>)]."
> > > ```
> > >
> > > **(4) Flexible Task Combination:** VisionLLM can also combine different tasks through instructions. For example, with the following instruction, we can combine localization and question-answering tasks to count the number of white cats.
> > >
> > > ```
> > > "Locate all the objects in the image that are part of the category set {'white cat': <c0>} and output their index of class label starting from 0 and offsets of bounding box coordinates. The bounding box should be a rectangle that covers the entire object. The offsets should be given as top-left and bottom-right corners of the rectangle relative to the center point and should be within <range>. The output format should be (c, x1, y1, x2, y2). The image is: <image>. <cls><x1><y1><x2><y2>...<cls><x1><y1><x2><y2>. How many white cats are in this image?"
> > > ```
> > >
> > > Despite demonstrating zero-shot capabilities in some unseen scenarios, VisionLLM has some limitations. Due to the limited scale of training data in this version, the connection between vision and language concepts still needs improvement, leading to hallucination issues in VQA and captioning tasks. Additionally, it also struggle with handling specialized terms in niche domains, such as "Magnetic Accelerator".

---

> > > > ### Comment · Reviewer_TifL · 2023-08-20
> > > > **Thank you for the clarification**
> > > >
> > > > The response is comprehensive and have addressed my concerns. I would encourage the authors to put these discussions into the revised version.

---

> > > > > ### Author Response · Authors · 2023-08-20
> > > > >
> > > > > We will add these points in the revised edition. Thanks for your efforts.

---

### Official Review · Reviewer_wdrB · 2023-07-06

**Soundness:** 3 good
**Presentation:** 3 good
**Contribution:** 3 good
**Rating:** 6
**Confidence:** 3

**Summary:**

This work proposes VisionLLM, a unified framework for vision tasks and vision-language tasks, using natural language task prompts. It demonstrates capabilities in a good variety of tasks.

**Strengths:**

- This is the first attempt to use natural language prompts for vision-centric tasks such as object detection and instance segmentation. I think this work is tackling an important problem.
- The language-guided image tokenizer is a novel component to convert image into tokens guided by text.
- VisionLLM treats image tokens the same way as text tokens, such that the entire task (including image and class list) can be encoded as one piece of text. This converts vision tasks into sequence generation problem handled by a LLM. (I'm actually not sure if this understanding is correct, so I ask this in the Questions section as well.)

**Weaknesses:**

The framework is not a simplistic one, consisting of various components and types of losses, e.g. using bipartite matching for one type of outputs. Representing image as a set of ${e_i, l_i}$ is also kind of specific.

The model can do VQA but I didn't see any VQA results in the paper.

Overall my main complaint is that the framework and pipeline seem a bit complicated, but it is designed to incorporate a wide variety of tasks and seems effective at it. It is a good first attempt at using natural language and LLM to tackle vision tasks.

**Questions:**

How are image tokens $T$ and `<text>` passed into the LLM? On line 166 it says `<image>` and `<question>` are placeholder tokens for image tokens and question tokens. I imagine `<question>` will be replaced by the actual question inside the language instruction text (`<text>`), but will the `<image>` token be replaced by the image tokens $T$? Figure 3 seems to imply that $T$ is passed into the LLM separately from `<text>`. I think it's more flexible if all inputs are formulated into one piece of text to pass into the LLM, as one can directly extend it to use more than one images, etc.

**Limitations:**

NA.

---

> ### Author Rebuttal · Authors · 2023-08-10
>
> **Q1: Pipeline seems a bit complicated.**
>
> **A1:** We would like to recap the components of VisionLLM. It has two key components: the language-guided image tokenizer and the LLM-based decoder. They work together in the following way:
>
> (1) Firstly, the visual encoder extracts features from the image at different scales, and the text encoder obtains features from the language input.
>
> (2) Then, in the language-guided image tokenizer, the multi-scale image features interact with the language features to generate language-guided image tokens. Each token is represented by embedding and location information.
>
> (3) Finally, these image tokens replace the placeholder `<image>` in the language prompts.  The resulting language prompts are fed into the LLM-based decoder for open-ended tasks.
>
> In addition, some specific designs, e.g., bipartite matching, representing images as a set of $(e_i, l_i)$, are useful to accelerate the convergence of VisionLLM. Generally speaking, all the components are closely interconnected and indispensable for VisionLLM. Building a generalist model for open-ended tasks is a complicated system engineering, and we will explore more simplified implementations in the future.
>
> **Q2: The model can do VQA but I didn't see any VQA results in the paper.**
>
> **A2:** For VQA tasks, we qualitatively showcase the performance of VisionLLM on complicated VQA scenarios, as shown in Figure 2(d) and Figure F in the supplementary material. There are two-fold reasons:
>
> (1) Our model was trained on the LLaVA-Instruct-150K dataset, which encourages the model to generate long and detailed answers for visual questions, as explained in L256-257. However, most existing VQA benchmarks expect short answers, which makes our model score low (VisionLLM-R50: 33.86 vqa-score on VQAv2 test-dev). It is unfair to compare our model with other models on these benchmarks.
>
> (2) As we discussed in the Q4 of Reviewer kyDb, the existing GPT-based evaluation methods (e.g., metrics in LLaVA) are unstable and are affected by the online closed-sourced model released by OpenAI. So there is currently no widely recognized standard for such VQA evaluation.
>
> **Q3: How are image tokens `<image>` and `<text>` passed into the LLM?**
>
> **A3:** The image tokens are directly placed at the placeholder `<image>` at the language prompt. We will make it clearer in our revised version.

---

> > ### Comment · Reviewer_wdrB · 2023-08-20
> > **Response**
> >
> > Thanks to the authors for the rebuttal. I maintain my opinion that the method is not simplistic enough, but I also maintain my positive rating of a weak accept.

---

> > > ### Author Response · Authors · 2023-08-20
> > >
> > > Thank you for your efforts in the review.

---

### Official Review · Reviewer_kyDb · 2023-07-07

**Soundness:** 3 good
**Presentation:** 2 fair
**Contribution:** 2 fair
**Rating:** 5
**Confidence:** 4

**Summary:**

This paper construction a vision/language model by passing visual features into a LLM. They trained the model on several standard tasks as well object detection and referring expression by adding special location tokens to the LLM's vocabulary. The model features a text-guided image tokenizer and an efficient decoding approach when generating segmentation or bounding boxes.

**Strengths:**

- The suggestion in section 3.4 seems like a nice way to avoid auto-regressive decoding when it is not needed, although the method was a bit hard to parse.
- Showing that the approach of adapting visual features for LLM can work objet detection and segmentation is interesting, these tasks have been less well explored in this area.
-  The quantitive examples in the appendix at least suggests the model has instruction following capabilities similar to other models tuned on the LLaVA dataset.

**Weaknesses:**

- The scores are decent but not amazing. Only slightly better than pix2seq if using the same backbone, and the CiDER scores are lower then relatively simple models like ClipClap or VL-T5.
- The central idea is essentially following the pix2seq method combined with the now pretty well studied method of adapting visual features for a LLM approach, which makes sense but does not feel hugely novel to me.
- The authors should consider evaluating follow the methodology of LLAVa given they are using that data.

**Questions:**

Did the authors try pre-training the model? Pre-training to initially learn the alignment for the vision/language component is common practice for these kinds of models.

Can the model generalize to outputting bounding boxes for tasks other than refexp or object detection? For example for pointing VQA questions where you output a bounding to answer a question.

In section 3.4, what does "...feed the tokens of structural output format as queries to the decoder" mean? That they are used as the initial starting token? What happens if there are multiple objects so they are then multiple class and x1 coordinates to produce? Or if the model needs to interleave text with structured output like in Figure 2a?

Table 1 is missing many models that achieve better scores, BEiT-3 is better at detection and captioning for example. I think they should be included for reference

**Limitations:**

I think the authors should at least note some of the potential issues with these kinds of models (bias, potential for abuse my generating misinformation, hallucination, ect.)

---

> ### Author Rebuttal · Authors · 2023-08-10
>
> **Q1: Details of the `output-format-as-query` decoding.**
>
> **A1:** We thank the reviewer's appreciation of our output-format-as-query design. In Common Questions Q1, we provide more details about output-format-as-query regarding the data construction, training, and inference process. Here, we answer your detailed questions about it:
>
> **Q1-1: What does "...feed the tokens of structural output format as queries to the decoder" mean? That they as the initial starting token?**
>
> **A1-1:** Yes, the parsed outputs are appended to the original user instructions, and the resulting instructions are then fed into the LLM-based decoder.
>
> **Q1-2: What happens if there are multiple objects so they are then multiple classes and x1 coordinates to produce? Or if the model needs to interleave text with structured output like in Figure 2a?**
>
> **A1-2:** For the perception tasks (e.g., detection), the output format consists of a string with 100 segments of "<cls><x1><y1><x2><y2>", which can accommodate multiple objects in the scene. For interleave text with structured output, our model naturally supports this, as constructed user instructions default to this format (see Common Questions Q1).
>
>
> **Q2: The scores are decent but not amazing.**
>
> **A2:** Besides performance scores, the primary objective of this work is to design a generalist vision model available for user-tailored open-ended tasks. However, there are a few issues that are beyond the scope of this paper:
>
> (1) Due to the shared weights of our generalist model, conflicts may arise among different tasks, potentially leading to lower performance compared to specialized or foundation models that follow the "pre-train then fine-tune" paradigm (e.g., Pix2Seq v1, BEiT-3);
>
> (2) We use LLaVA-Instruct-150K to preserve the language capability of LLMs during training, which tends to generate long and detailed captions. But these long captions are suboptimal for traditional BLEU and CIDEr metrics.
>
>
> **Q3: The central idea is essentially following the pix2seq method.**
>
> **A3:** We argue that our model significantly differs from Pix2Seq v1/v2 in terms of its ability to handle open-ended tasks, model architecture, and decoding process (See Common Questions Q2). We think the reason why the two models may seem similar is that both our model and the Pix2Seq series use coordinate discretization to model the perception task, but this is NOT the main contribution of this work.
>
> **Q4: The authors should consider evaluating following the methodology of LLaVA given they are using that data.**
>
> **A4:** The evaluating method in LLaVA is unstable as it is affected by the online version of GPT-4, which is a closed-source system and not available in all countries or regions. Furthermore, there is no widely recognized average standard for such evaluation at the moment. Therefore, we adopt a more stable and controllable evaluation approach in this work. We test the performance of VisionLLM by generalizing it to various representative visual perception and understanding tasks, and evaluate it with standard benchmarks. In addition, we also design variant evaluations based on standard benchmarks to examine the open-ended task ability of our models (see Table 2(a)(b), and Figure B and C in supplementary material).
>
> **Q5: Did the authors try pre-training the model? Pre-training to initially learn the alignment for the vision/language component is common practice for these kinds of models.**
>
> **A5:** The first training stage of VisionLLM involves open-ended detection tasks with random user instructions. This is a specific form of vision-language alignment.
>
> **Q6: Can the model be generalized to output bounding boxes for tasks other than refexp or object detection? For example, for pointing VQA questions where you output a bounding to answer a question.**
>
> **A6:** Yes, our model can be generalized to use bounding boxes to answer questions. For example, as shown in Figure 2(a), when you ask a question like "What is the child eating?" in the class set `<class>`, VisionLLM will predict the bounding box of the doughnut as the answer to this question. We present more examples in Figure I in the attached PDF file for rebuttal to show this feature of VisionLLM.
>
> **Q7: Table 1 missing many models that achieve better scores, BEiT-3 is better at detection and captioning for example. I think they should be included for reference.**
>
> **A7:** In Table 1, we have listed recently popular generalist models capable of handling various tasks using shared weights. Differently, BEiT-3 is a foundational model that utilizes additional decoders for fine-tuning, incorporating a range of specialized designs, which do not support open-ended tasks. We will add and discuss these works in our revised version. Thanks for your suggestion.
>
> **Q8: I think the authors should at least note some of the potential issues with these kinds of models (bias, potential for abuse by generating misinformation, hallucination, etc.**
>
> **A8:** Thank you for the suggestions! We will attempt to address this through some de-biasing methods before the model is released.

---

### Author Rebuttal · Authors · 2023-08-10

Dear all reviewers:

We sincerely appreciate the reviewers for their time and effort in the review. This submission received 5 review comments, and 4 reviewers gave positive scores. We first address some common questions, followed by detailed responses to each reviewer separately. We hope our responses could clarify existing doubts.

&emsp;

###  Common Questions
**Q1: Details of the `output-format-as-query` decoding.**

**A1:** The `output-format-as-query` decoding technique is designed to parse the standard output format, which is compatible with the LLM-based decoder, from user instructions. The details are as follows:

**Data Construction:** Following self-instruct [1], we create various user instructions for each task to simulate human interaction. Here are some examples:
```
System message:
"You are an AI assistant for translating the user instructions to the standard prompt. Please help me parse the following input.
Input: {input}
Output:"

# Object detection
Input:
"The image is: <image>. Please thoroughly examine the image and detect all objects belonging to the category set {'person': <c0>, 'bicycle': <c1>, 'car': <c2>, 'motorcycle': <c3>}."
Output:
"The bounding boxes are <cls><x1><y1><x2><y2><cls><x1><y1><x2><y2>...<cls><x1><y1><x2><y2>."

# Image caption
Input:
"The image is: <image>. Please write a short caption for this image."
Output:
"The image shows that <bos>"
```

**Training:** After obtaining the data of user instructions, we finetune Alpaca using the next token prediction task for supervision, making it able to accomplish the output format parsing process.

**Inference:** As described in L229-236 and Figure 4, the inference process involves the following steps:

(1) We first use the fine-tuned Alpaca to parse the user instructions into standard output formats for different tasks. For instance, in the case of object detection, the output format may be "The bounding boxes are <cls><x1><y1><x2><y2><cls><x1><y1><x2><y2>...<cls><x1><y1><x2><y2>". For image captioning, the output format could be "The image shows that <bos>".

(2) The parsed outputs are then appended to the original user instructions as suffix texts. The extended instructions are fed into the LLM-based decoder as queries.

(3) Since the output format contains special tokens, such as <cls>, <x1>, <y1>, <x2>, <y2>, and <bos>, by treating these tokens as queries, the LLM-based decoder can predict the corresponding results. This approach enables the detection task to run in parallel like the cloze task, while the captioning task remains the next token prediction.

We will make it clearer in our revised version.

&emsp;

**Q2: The difference between VisionLLM and Pix2Seq.**

**A2:** Although both VisionLLM and Pix2Seq v1/v2 employ coordinate discretization for object detection tasks, _**they differ significantly in terms of task generality, model design, and decoding process**_.

**Task Generality:**  VisionLLM allows users to customize vision tasks using language instructions, supporting user-tailored output formats, task targets, task descriptions, etc.  In contrast, Pix2Seq v1 is a special model for object detection, and  Pix2Seq v2 only supports pre-defined task switching with learnable prompt tokens, lacking the flexibility of task customization.

**Model Design:** VisionLLM consists of a series of careful designs for open-ended tasks, including (1) language instructions that align vision tasks with NLP tasks; (2) a flexible tokenizer guided by natural language instructions (Pix2Seq v2 uses _**unreadable embedding**_ for task switching); and (3) an open-ended task decoder based on LLMs along with an improved decoding process.

**Decoding Process:** Pix2Seq struggles to converge in open-ended task scenarios with random user instructions (see Table A(d) in supplementary material). VisionLLM solves this problem effectively by using its output-format-as-query approach, which enables the model to work with the Hungarian matching loss and handle highly random open-ended task instructions efficiently.

To sum up, VisionLLM and Pix2Seq are distinct models. Pix2Seq is a pioneering generalist model but has limitations (see Figure 1(a)). VisionLLM explores new possibilities for end-to-end models that unify vision and language tasks in the LLM era. We will clarify this in the revised version.

&emsp;

**Q3: Evaluation VisionLLM on more backbones.**

**A3:** We chose to evaluate our model on ResNet-50 and InternImage-H backbones because ResNet-50 is widely recognized as a representative backbone at a common scale, and InternImage-H is known for its large-scale backbone with top-notch performance. Results on ResNet-50 and InternImage-H demonstrate the generality of VisionLLM on backbones at different parameter scales. In the table below, we have included the results of ViT-B, which still meet our expectations.

| Method | Backbone | Open-Ended |  | Detection |  |  | Instance Seg. |  | Grounding | Captioning |   |
| - | - | - | - | - | - | - | - | - | - | - | - |
|  |  |  | AP | AP50 | AP75 | AP | AP50 | AP75 | P\@0.5 | BLEU-4 | CIDEr |
| Uni-Perceiver | ViT-B | - | - | - | - | - | - | - | - | 32.0 | - |
| Uni-Perceiver-MoE | ViT-B | - | - | - | - | - | - | - | - | 33.2 | - |
| Uni-Perceiver-V2  | Swin-B | -  | 58.6 | - | - | 50.6 | - | -  | - | 35.4 | 116.9 |
| Pix2Seq v2 | ViT-B | - | 46.5 | -  | - | 38.2 | - | - | - | 34.9 | - |
| VisionLLM-R50 | ResNet-50 | ✓  | 44.6 | 64.0  | 48.1 | 25.1 | 50.0 | 22.4 | 80.6  | 31.0 | 112.5 |
| VisionLLM-ViT-B | ViT-B | ✓ | 47.3 | 68.6 | 51.4 | 26.8 | 57.7 | 22.6 | 81.3  | 31.5  | 113.1 |
| VIsionLLM-H | Intern-H | ✓ | 60.2 | 79.3 | 65.8 | 30.6 | 61.2 | 27.6 | 86.7  | 32.1 | 114.2 |


[1] Wang, Yizhong, et al. "Self-instruct: Aligning language model with self-generated instructions." arXiv preprint arXiv:2212.10560 (2022).

---

### Decision · Program_Chairs · 2023-09-21

**Decision:**

Accept (poster)

**Comment:**

While initially there were mixed ratings, there was an effective rebuttal and five knowledgeable reviewers recommended accepting the paper: Weak Accept, Accept, Weak Accept, Borderline Accept, and Borderline Accept.  The paper presents an effective framework that provides a unified perspective for vision and language tasks treating images as a visual language and aligning vision-centric tasks. The authors build on top of existing models but extend them to facilitate predicting boxes which improves the explainability perspective and augments foundation models with grounding capabilities.  Based on the clear positive consensus, it is our recommendation to accept the paper. No basis to overturn the reviews.  Authors should attend to the main points in the reviews. when preparing a final version. No basis to overturn the reviews.